# The Surprising Effectiveness of Membership Inference with Simple N-Gram Coverage

**Skyler Hallinan**[♠], **Jaehun Jung**[♡], **Melanie Sclar**[♡],
**Ximing Lu**[♡], **Abhilasha Ravichander**[♡], **Sahana Ramnath**[♠],
**Yejin Choi**[♣], **Sai Praneeth Karimireddy**[♠], **Niloofar Mireshghallah**[♡], **Xiang Ren**[♠]
[♠]University of Southern California    [♡]University of Washington
[♣]Stanford University
shallina@usc.edu

## Abstract

Membership inference attacks serves as useful tool for fair use of language models, such as detecting potential copyright infringement and auditing data leakage. However, many current state-of-the-art attacks require access to models' hidden states or probability distribution, which prevents investigation into more widely-used, API-access only models like GPT-4. In this work, we introduce N-GRAM COVERAGE ATTACK, a membership inference attack that relies **solely** on text outputs from the target model, enabling attacks on completely black-box models. We leverage the observation that models are more likely to memorize and subsequently generate text patterns that were commonly observed in their training data. Specifically, to make a prediction on a candidate member, N-GRAM COVERAGE ATTACK first obtains multiple model generations conditioned on a prefix of the candidate. It then uses n-gram overlap metrics to compute and aggregate the similarities of these outputs with the ground truth suffix; high similarities indicate likely membership. We first demonstrate on a diverse set of existing benchmarks that N-GRAM COVERAGE ATTACK outperforms other black-box methods while also impressively achieving comparable or even better performance to state-of-the-art white-box attacks – despite having access to only text outputs. Interestingly, we find that the success rate of our method scales with the attack compute budget – as we increase the number of sequences generated from the target model conditioned on the prefix, attack performance tends to improve. Having verified the accuracy of our method, we use it to investigate previously unstudied closed OpenAI models on multiple domains. We find that more recent models, such as GPT-4o, exhibit increased robustness to membership inference, suggesting an evolving trend toward improved privacy protections[1].

## 1  Introduction

While training data serves a central role in developing modern large language models, model providers have increasingly withhold critical details of their datasets (Brown et al., 2020; Touvron et al., 2023b; Jiang et al., 2023). The lack of data provenance is particularly problematic, as models are often exposed to copyrighted data such as novels during training (Henderson et al., 2023a; Carlini et al., 2022), which they may regurgitate in their generations post-deployment (Chen et al., 2024; Biderman et al., 2023a). This has led to multiple lawsuits from news providers like the New York Times, who assert that these model tendencies decrease the utility of their protected works (Grynbaum & Mac, 2023; Bruell, 2025).

Membership inference attacks, methods to posit whether or not specific text documents were in the training data of some model, are increasingly common strategies to audit the training

---

[1]We release our code and data at https://github.com/shallinan1/NGramCoverageAttack

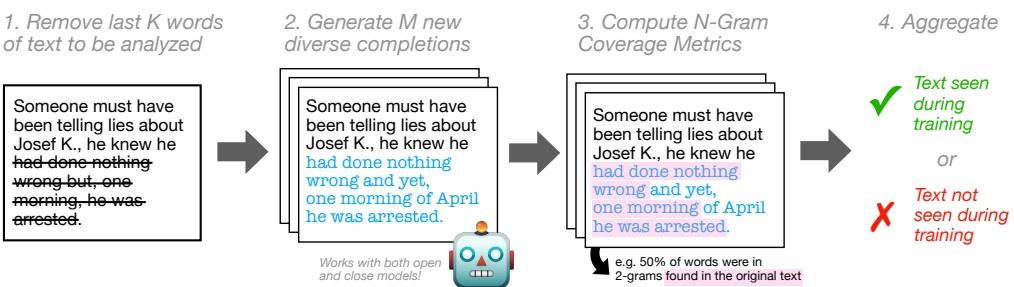

Figure 1: A high-level overview of N-GRAM COVERAGE ATTACK, a cost-effective, white-box membership inference attack effective for both open and closed models: (1) attain a short prefix of the candidate document (2) sample continuations given the prefix from the target model (3) compare to the original suffix (4) aggregate similarities to infer membership.

data of large language models (Carlini et al., 2020). However, many current methods require access to the underlying model or logits (Yeom et al., 2017; Mattern et al., 2023; Shi et al., 2023), limiting their scope to a narrow set of models. Notably, this excludes larger, more capable, and more popular models like GPT-4 (Achiam et al., 2023) which restrict access to this information, outputting only model generations.

In this work, we investigate whether membership inference attack can be done with only the access to sampled model outputs, and if so, whether such an approach can perform comparable to established white-box methods. We introduce **N-GRAM COVERAGE ATTACK**, a family of membership inference attacks based on only the surface-form similarity of model generations against the input document. Our approach, illustrated in Figure 1, is cost-effective and applies to black-box models. It involves three steps: (1) sample multiple reconstructions of the input document given a short prefix (2) measure the similarity of each reconstruction against the original document's suffix, and (3) aggregate the similarities to infer membership.

Our analyses reveal that N-GRAM COVERAGE ATTACK substantially simplifies previous MIA methods in the black-box setting (Duarte et al., 2024), yet surprisingly performs on-par or even better than white-box methods that require model loss or token logits. We also explore the effectiveness of our approach on models' *post-training data*, an area under-explored in the literature. Compared to recent membership inference methods for black-box models (Duarte et al., 2024), our method is substantially compute-efficient (refer to specific numbers), and importantly, scales better by increasing the size of repeated sampling. We additionally collect and test our method on two new datasets for membership inference and benchmark many new models such as TÜLU. Having verified the accuracy of our method, we use it to investigate previously unstudied closed OpenAI models on multiple domains. We find that more recent models, such as GPT-4o, exhibit increased robustness to membership inference, suggesting an evolving trend toward improved privacy protections.

## 2 Background and Related Work

In this work, we propose a membership inference method that can work on both closed, black-box models, and on open-weight models. We briefly summarize prior efforts in membership inference, memorization, training data extraction, and techniques for protection.

**Membership Inference** Tracing member data was first proposed in the context of genomic privacy (Homer et al., 2008; Sankararaman et al., 2009) before later being explored for deep neural networks (Shokri et al., 2016). For large language models, most previous work utilize the prediction loss of a candidate sequence, with the intuition that models are likely to have a lower loss on sequences that have been seen during training. Yeom et al. (2017) use a simple threshold – the loss itself – while Carlini et al. (2020) additionally use a *reference model*, a language model with less memorization, to remove the effect of the intrinsic sequence

difficulty from the observed loss. While this technique has found widespread use (Mattern et al., 2023; Mireshghallah et al., 2022; Ye et al., 2021; Fu et al., 2024), it is difficult to ascertain whether the reference model itself has memorized the sequence. Carlini et al. (2020) instead normalize sequences by their *compressed size* (entropy) via the zlib library, while Shi et al. (2023) propose Min-K%, which uses the log-likelihood of the $K\%$ most unlikely tokens as a membership signal. Zhang et al. (2024b) later extend this by leveraging statistics from the entire vocabulary distribution to normalize token probabilities; the key intuition is changing the membership signal from absolute to relative token probabilities.

However, none of these methods work with only model outputs. Fu et al. (2025) demonstrate that fine-tuning large language models can enable effective membership inference detection; however, this approach assumes that the model provider permits fine-tuning, besides requiring supervision. Duarte et al. (2024) introduce an output-only attack which formulates membership inference as a question-answering task: it *paraphrases* candidate documents, then tests model preferences by presenting them alongside the ground truth; if a candidate was truly trained on, its paraphrases are more likely to be favored above chance. Hisamoto et al. (2020) demonstrate that n-gram features are effective for membership inference in sequence-to-sequence models for machine translation.

**Memorization** There has also been work aimed at identifying *memorization* of training data in black-box large language models (Carlini et al., 2022). These output-only approaches typically either examine whether models can produce verbatim continuations for an input sequence (Karamolegkou et al., 2023; Zhao et al., 2024; Freeman et al., 2024; Henderson et al., 2023b), or if models can reproduce certain tokens in documents that are difficult without memorization (Chang et al., 2023b; Ravichander et al., 2025). Zhang et al. (2024a) even argue that such approaches could provide training data proofs with controlled false-positive rates. However, these works typically only focus on identifying a subset of member data that models can reproduce with high fidelity, whereas membership inference methods like ours focus on a stricter regime: the distinguishability of all member and non-member data. There have also been several efforts that aim to uncover memorization evidence, assuming access to the model's prediction loss over a sequence (Garg et al., 2024; Ravikumar et al., 2024). In contrast, our work is based on the fully black-box setting, where we only assume API-level access to the model.

Considerable prior efforts have also focused on the extraction setting for memorization, aiming to reveal training data directly from model generations rather than via membership queries (Carlini et al., 2021; Nasr et al., 2023a; Bai et al., 2024); our work differs as we aim to determine the membership of any given input rather than just ones that can be elicited from the model. Considerable work has also sought to prevent the success of any attack to extract or identify training data from large language models (Siyan et al., 2025; Tang et al., 2021; Jia et al., 2019). Our work contributes to this growing body of literature by providing a previously unknown approach to identify membership in large language models.

**Protection Methods** Privacy-preserving training algorithms such as DP-SGD (Abadi et al., 2016) provide provable guarantees about the extent of possible memorization. Post-training methods such as unlearning (Cao & Yang, 2015) can also be used to certifiably "forget" problematic data which was memorized (Sekhari et al., 2021). However, such theoretical guarantees can be overly conservative and demand sacrificing too much utility. Instead, empirical *privacy auditing* methods (Jagielski et al., 2020; Nasr et al., 2023b; Steinke et al., 2023), which rely on membership inference attacks, form the basis of most production privacy evaluations (Song & Marn, 2020). This makes designing reliable and consistent membership inference techniques critical for privacy evaluations.

## 3 Method

In this section, we introduce N-GRAM COVERAGE ATTACK, a cost-effective method for membership inference that only requires model-generated samples without relying on any model internals like token logits. Below, we formalize the membership inference task (§3.1), illustrate our framework that leverages n-gram statistics (§3.2), then discuss the design of scoring function variants that comprises the family of N-GRAM COVERAGE ATTACK.

---

**Algorithm 1** Membership Inference with N-GRAM COVERAGE ATTACK

---

**Require:** Target model $M_\theta$, input text $x$ to test for membership, threshold $\epsilon$, token index $k$, number of outputs to sample $d$, similarity function $\texttt{sim}$

**Ensure:** Prediction: **Member** or **Non-member**

1: **Sample** $d$ generations from model $M_\theta$ using part of $x$ as the prompt:
$$\{o_\theta^{(i)}\}_{i=1}^d \leftarrow M_\theta(x_{\leq k})$$

2: Compute the **similarities** of the generations to the suffix of $x$ unseen in step 1:
$$S_\theta^{(i)} \leftarrow \texttt{sim}\left(o_\theta^{(i)}, x_{>k}\right), \forall i = 1, \ldots, d$$

3: **Aggregate** the similarities using $\text{agg}(x)$:
$$S_\theta^{\text{agg}} \leftarrow \text{agg}\left(\{S_\theta^{(i)}\}_{i=1}^d\right)$$

4: Predict **Member if** $S_\theta^{\text{agg}} > \epsilon$ **else** Predict **Non-member**

---

### 3.1 Membership Inference Task

A language model $M_\theta$ is trained on a collection of data $\mathcal{D}$, where each sample $x^+ \in \mathcal{D}$ denotes a **member**, and $x^- \notin \mathcal{D}$ denotes a **non-member**. Given some target model $M_\theta$ and a corpus of *candidate* text documents $\mathcal{C}$, a **membership inference attack** attempts to determine $\mathcal{C} \cap \mathcal{D}$: which, if any, samples $x \in \mathcal{C}$ were used in the training of $M_\theta$.

### 3.2 N-GRAM COVERAGE ATTACK: Membership Inference using only Model Outputs

The goal of our algorithm is to assess if a model $M_\theta$ has likely been trained on a particular sequence $x$. We achieve this through approximating how $M_\theta$ has memorized a specific sequence $x$ by *empirically* measuring how closely the model's sampled outputs align with that sequence. Our key intuition is that models should output text that is more similar to data that they were trained on (member data) than data they were not trained on (non-member data) (Carlini et al., 2022; Chen et al., 2024). Specifically, we prompt the model multiple times with a prefix of $x$ and assess how close its outputs are to naturally "regenerating" the remaining suffix of $x$.

Formally, N-GRAM COVERAGE ATTACK consists of three steps, detailed below and in Algorithm 1. First, given some prompt $p$ and a prefix of $x$ as input of size $k$, $x_{\leq k}$, we sample $d$ diverse completions with $M_\theta$ using standard language modeling (**Sample from Target Model**). $p$ will usually contain an instruction prompt to reconstruct text.

$$\left\{o_\theta^{(i)}\right\}_{i=1}^d \sim M_\theta(\cdot | p, x_{\leq k})$$

We then assess the similarity of the sampled generations $o_\theta^{(i)}$ to the original suffix of $x$, $x_{>k}$, where $\texttt{sim}(x_1, x_2)$ computes the *similarity* between two texts $x_1$ and $x_2$ (higher is better) **(Compute Similarities of Outputs with Original Document)**:

$$S_\theta^{(i)} \leftarrow \texttt{sim}(o_\theta^{(i)}, x_{>k}), \quad \forall i = \{1, \ldots, d\}$$

Finally, we condense the $d$-dimensional vector of scores into a single value using an aggregation function $\text{agg}(x) : \mathbb{R}^d \to \mathbb{R}$. We predict $x$ to be a **member** if and only if $\text{agg}(x) > \epsilon$ for some pre-defined value $\epsilon$ (**Aggregate**).

### 3.3 N-GRAM COVERAGE ATTACK Function Choices

Our method naturally allows for using different similarity metrics $\texttt{sim}(\cdot)$ and aggregation functions $\text{agg}(\cdot)$ depending on the use-case; we detail these below.

**Similarity Metrics**  We consider three distinct $\texttt{sim}(x_1, x_2)$ function options to be used with N-GRAM COVERAGE ATTACK: Coverage, Creativity Index (Lu et al., 2024), and Longest

Common Substring (LCS) Notably, **these are all simple, n-gram coverage metrics**, which are both interpretable and efficient to compute.

Coverage (Cov) quantifies the overlap between two documents $x_1$ and $x_2$ by computing the proportion of tokens in $x_2$ covered by matching $n$-grams of at least length $L$ from $x_1$.

$$\text{Cov}_L(\mathbf{x_1}, \mathbf{x_2}) = \frac{\sum_{w \in \mathbf{x_2}} \mathbb{1}\big(\exists \ n\text{-gram } \mathbf{g} \subseteq \mathbf{x_1}, \|\mathbf{g}\| \geq L \ \text{ s.t. } w \in \mathbf{g}\big)}{\|\mathbf{x_2}\|} \in [0, 1]$$

Creativity Index (Cre; Lu et al., 2024) measures textual novelty by penalizing repeated content from reference materials at multiple N-gram lengths. It sums 1 - coverage over increasing N-gram sizes, rewarding texts with lower and shorter-span overlaps:

$$\text{Creativity Index}(\mathbf{x_1}, \mathbf{x_2}) = \sum_{L=A}^{B} 1 - \text{Cov}_L(\mathbf{x_1}, \mathbf{x_2}) \in [0, B - A]$$

In practice, we use $-$Creativity Index so higher scores indicate more similarity.

Longest Common Substring (LCS) computes the length of the longest common contiguous substring between $x_1$ and $x_2$. This can be done on multiple granularities, such as on the character or word level ($LCS_C$ and $LCS_W$). Unlike coverage and creativity index, we do not include length normalization here.

**Aggregation Function** We consider four simple $\text{agg}(x)$ functions: maximum, minimum, mean and median. Since false positives – true non-members which can be accurately reproduced by the model – are unlikely, the max metric is particularly appealing, as it effectively surfaces the strongest membership signals, even if they are sparse.

## 4 Experiments

We perform comprehensive experiments across four datasets, multiple model families across scale, and several baselines, demonstrating that N-GRAM COVERAGE ATTACK is a versatile and effective attack, despite its simplicity. For all experiments, we perform an initial sweep with a small 5% validation set to finalize hyperparameters before reporting test set results. See Appendix §B for more details and §A for additional results on the Pile and Dolma.

### 4.1 Models

We consider a diverse set of models to attack, which vary in size and model access. For open-weight models, we include **LLaMa 1** (Touvron et al., 2023a), a set of decoder-only language models with sizes of 7B, 13B, 30B, and 65B respectively released by Meta in February 2023.

We also include a large suite of closed, API-access OpenAI models offering only access to output texts[2], largely understudied for membership inference. We start with **GPT-3.5 Instruct** (gpt-3.5-turbo-instruct) (Brown et al., 2020), designed to replace the now-deprecated text-davinci-003, OpenAI's first instruction-following model (Ouyang et al., 2022). We also include two **GPT-3.5 Turbo** (OpenAI, 2022) models, the first set of chat-specific OpenAI language models: gpt-3.5-turbo-0125 and gpt-3.5-turbo-1106. All three GPT-3.5 models have a knowledge cutoff of Aug 31, 2021. We also consider **GPT-4 Turbo** (Achiam et al., 2023) the follow-up to GPT-3.5 Turbo, **GPT-4o** (OpenAI, 2024), a contemporary, flagship OpenAI model released in mid-2024, and **GPT-4o mini** (OpenAI, 2024), the cost-efficient, smaller version released shortly after. These all have a cutoff date of late 2023.

Finally, we also consider **TÜLU** (Wang et al., 2023), a suite of varying-scale, models converted from base to instruction-tuned variants by training on a curated human + machine-generated data mixture. We use TÜLU 1 (base models of LLaMa 7B, 13B, 30B, and 65B (Touvron et al., 2023a) and TÜLU 1.1 (base models of LLaMA-2 7B, 13B, and 70B (Touvron et al., 2023b)).

---

[2]Some output limited output text probabilities, but they are not used by existing baselines

## 4.2 Datasets

We use a set of five diverse datasets — three existing, and two we construct — to comprehensively evaluate membership inference attacks. While most of the datasets are used to assess pretraining membership, we also construct a new dataset to assess *fine-tuning* membership. See Appendix §D for more details on all datasets and our data creation procedure.

**BookMIA** (Shi et al., 2023) consists of 512-word snippets sampled from 100 books. Half of the data comes from famous literature presumed to be in the training corpus of older OpenAI models like GPT-3.5 (Chang et al., 2023a). The other half is comprised of books published *after* 2023. We use the GPT-3.5 family as the target models.

**WikiMIA** (Shi et al., 2023) consists of snippets from Wikipedia articles written before 2017 and articles written after 2023; for models released in this time span, these are members and non-members respectively [3]. Following prior work (Shi et al., 2023), we use base LLaMa 7B, 13B, 30B and 65B (Touvron et al., 2023a) and GPT-3.5 as the target models.

**WikiMIA$_{2024}$ Hard** is a new dataset we construct which builds upon the original WikiMIA format with two key modifications for more robust evaluation. (1) First, to minimize temporal distributional differences between members and non-members, we identify Wikipedia summaries whose content has changed from their version at the end of 2016 to versions updated in 2024 or later. Following WikiMIA's core assumption, we treat pre-2017 summary versions as likely members of model training sets, as these were presumably scraped into massive pretraining corpora, while non-members are the most recent versions of these same summaries, edited after most models' knowledge cutoff dates. By using different versions of the same articles, we minimize topical differences between members and non-members, unlike the original WikiMIA, where members and non-members cover entirely different topics and time periods. (2) Second, our target models include not only GPT-3.5 and the LLaMa family (as in the original WikiMIA), but now also extend to more recent models such as GPT-4o and GPT-4, which have knowledge cutoff dates near the end of 2023.

We also note that we filter article pairs for a minimum edit distance (Levenshtein, 1965), ensuring the newer (non-member) version differs meaningfully from the older (member) one. This makes the benchmark challenging but not impossible, so that observed model performance reflects actual memorization capability rather than the inability to detect imperceptible differences. Finally, we also constrain length variations between versions to within 20% to avoid spurious length features in members and non-members.

**WikiMIA-24** (Fu et al., 2025) follows the original WikiMIA (Shi et al., 2023) collection methodology with an updated cutoff for non-members; members are still Wikipedia articles written before 2017, while non-members are now articles written after March 1, 2024[4]. The target models are the same as WikiMIA$_{2024}$ Hard.

**TÜLU Mix** (Wang et al., 2023) is a new membership dataset we construct to assess *fine-tuning* membership attack effectiveness. Since most previous work investigates pre-training membership, we seek to understand how well existing strategies transfer to fine-tuning. The TÜLU Mix was used to train both the TÜLU 1 and 1.1 suite of models, which are the natural target models. The authors test a variety of candidate instruction-tuning datasets across domains, unifying the format before selecting a subset as the best mixture. We use these data points as members and data from the instruction datasets not-selected as non-members.

## 4.3 Baselines

We detail the baselines we run for all the membership inference tasks:

---

[3]As Duan et al. (2024) note, this collection methodology may result in spurious, temporal distribution shift between members and non-members. However, given that these are among the only available benchmarks that can be used for closed-access OpenAI models (due to a lack of publicly-known training data) (Shi et al., 2023), we believe it is important to include them, even with their known limitations, to enable broader evaluation and comparison.

[4]As WikiMIA-24 only updates the temporal boundary without modifying the underlying data collection procedure, it likely inherits the same temporal distribution shift vulnerabilities as in WikiMIA

| Model | N-GRAM COVERAGE ATTACK | | | | | White-Box Attacks | | | |
|---|---|---|---|---|---|---|---|---|---|
| | Cov. | Cre. | LCS$_c$ | LCS$_w$ | D-C | Loss | R-Loss | zlib | MinK |
| **WikiMIA (Shi et al., 2023)** | | | | | | | | | |
| GPT-3.5-0125 | **0.64** | 0.63 | 0.61 | 0.60 | 0.55 | - | - | - | - |
| GPT-3.5 Inst. | **0.62** | 0.61 | 0.58 | 0.58 | 0.54 | - | - | - | - |
| GPT-3.5-1106 | **0.64** | 0.62 | 0.61 | 0.60 | 0.52 | - | - | - | - |
| LLaMa-7B | **0.60** | 0.59 | 0.56 | 0.55 | 0.48 | 0.62 | - | 0.63 | 0.64 |
| LLaMa-13B | **0.62** | 0.59 | 0.57 | 0.54 | 0.52 | 0.64 | 0.63 | 0.65 | 0.66 |
| LLaMa-30B | **0.63** | 0 62 | 0.57 | 0.58 | 0.49 | 0.66 | 0.69 | 0.67 | 0.69 |
| LLaMa-65B | **0.65** | 0.64 | 0.61 | 0.58 | 0.50 | 0.68 | 0.74 | 0.69 | 0.70 |
| **WikiMIA-24 (Fu et al., 2025)** | | | | | | | | | |
| GPT-3.5-0125 | **0.67** | **0.67** | 0.64 | 0.66 | 0.48 | - | - | - | - |
| GPT-3.5 Inst. | **0.65** | 0.64 | 0.62 | 0.64 | 0.50 | - | - | - | - |
| GPT-3.5-1106 | **0.68** | 0.67 | 0.66 | 0.68 | 0.49 | - | - | - | - |
| GPT-4 | **0.84** | 0.82 | 0.76 | 0.79 | 0.56 | - | - | - | - |
| GPT-4o-1120 | **0.83** | 0.82 | 0.77 | 0.79 | 0.50 | - | - | - | - |
| GPT-4o Mini | 0.73 | **0.74** | 0.66 | 0.69 | 0.44 | - | - | - | - |
| LLaMA-7B | 0.59 | 0.59 | **0.60** | 0.59 | 0.53 | 0.67 | - | 0.67 | 0.69 |
| LLaMA-13B | **0.63** | **0.63** | 0.61 | 0.61 | 0.50 | 0.68 | 0.60 | 0.69 | 0.71 |
| LLaMA-30B | **0.67** | 0.66 | 0.64 | 0.64 | 0.48 | 0.72 | 0.69 | 0.72 | 0.74 |
| LLaMA-65B | 0.64 | **0.65** | **0.65** | **0.65** | 0.50 | 0.74 | 0.74 | 0.75 | 0.76 |
| **WikiMIA$_{2024}$ Hard** | | | | | | | | | |
| GPT-3.5-0125 | **0.59** | 0.56 | 0.54 | 0.55 | 0.47 | - | - | - | - |
| GPT-3.5 Inst. | **0.64** | 0.63 | 0.61 | 0.61 | 0.45 | | - | - | - |
| GPT-3.5-1106 | **0.58** | 0.58 | 0.56 | 0.57 | 0.49 | - | - | - | - |
| GPT-4 | 0.57 | **0.58** | 0.55 | 0.57 | 0.44 | - | - | - | - |
| GPT-4o-1120 | **0.55** | 0.55 | 0.54 | 0.52 | 0.51 | - | - | - | - |
| GPT-4o Mini | **0.55** | 0.53 | 0.52 | 0.51 | 0.43 | - | - | - | - |
| LLaMa-7B | **0.55** | 0.54 | 0.53 | 0.52 | 0.47 | 0.51 | - | 0.50 | 0.52 |
| LLaMa-13B | **0.59** | 0.58 | 0.53 | 0.53 | 0.51 | 0.53 | 0.57 | 0.51 | 0.54 |
| LLaMa-30B | **0.61** | **0.61** | 0.55 | 0.57 | 0.50 | 0.56 | 0.61 | 0.53 | 0.60 |
| LLaMa-65B | **0.64** | 0.63 | 0.59 | 0.60 | 0.51 | 0.57 | 0.57 | 0.54 | 0.58 |

Table 1: Results for different models and attacks on WikiMIA, WikiMIA-24, and WikiMIA$_{2024}$ Hard. **Bold** denotes the best performance in the black-box attacks, while underline denote the best performance for the white-box attacks. The columns in blue are from N-GRAM COVERAGE ATTACK, while the columns in gray are loss-based baselines as a reference.

**Loss** (Yeom et al., 2017) uses the likelihood of a candidate member under the target model as a proxy for membership; higher likelihood (lower loss) samples are likely members.

**Reference Loss** (**R-loss**; Carlini et al. 2020) builds on the naive loss by subtracting the loss from a *reference* model – a smaller language model with general language ability and minimal memorization – to identify loss differences due to memorization rather than fluency.

**zlib** (Carlini et al., 2020) divides the loss by the compressed file size of the candidate member using the zlb library. The idea is that more compressible sequences – typically those with higher redundancy or lower entropy – should naturally have lower loss,

**Min-K%** (Shi et al., 2023) measures the likelihood of the $k$% least-likely tokens (*outlier* tokens) in the given text under the target model.

**DE-COP** (**D-C**; Duarte et al. 2024) formulates membership inference as question-answering task, where a model is prompted to infer the plausible completion to the input text.

| Model | N-Gram Coverage Attack | | | | | White-Box Attacks | | | |
|---|---|---|---|---|---|---|---|---|---|
| | Cov. | Cre. | LCS$_c$ | LCS$_w$ | D-C | Loss | R-Loss | zlib | MinK |
| GPT-3.5-0125 | 0.84 | **0.85** | 0.84 | 0.83 | 0.84 | - | - | - | - |
| GPT-3.5 Inst. | 0.91 | 0.91 | 0.92 | **0.93** | 0.68 | - | - | - | - |
| GPT-3.5-1106 | 0.84 | **0.85** | 0.83 | 0.84 | **0.85** | - | - | - | - |

Table 2: Results for BookMIA. **Bold** denotes the best performance in the black-box attacks. The columns in gray are white-box baselines **which cannot be computed** for these models.

| Model | N-Gram Coverage Attack | | | | | White-Box Attacks | | | |
|---|---|---|---|---|---|---|---|---|---|
| | Cov. | Cre. | LCS$_c$ | LCS$_w$ | D-C | Loss | R-Loss | zlib | MinK |
| TÜLU-7B | **0.79** | **0.79** | 0.73 | 0.74 | 0.48 | 0.84 | - | 0.81 | 0.84 |
| TÜLU-13B | **0.80** | **0.80** | 0.74 | 0.76 | 0.47 | 0.87 | 0.63 | 0.83 | 0.87 |
| TÜLU-30B | **0.82** | **0.82** | 0.76 | 0.77 | 0.52 | 0.87 | 0.54 | 0.84 | 0.87 |
| TÜLU-65B | 0.85 | **0.86** | 0.80 | 0.80 | 0.45 | 0.92 | 0.68 | 0.90 | 0.92 |
| TÜLU-1.1-7B | 0.72 | **0.73** | 0.70 | 0.71 | 0.47 | 0.77 | - | 0.74 | 0.76 |
| TÜLU-1.1-13B | **0.76** | 0.75 | 0.71 | 0.72 | 0.43 | 0.81 | 0.58 | 0.78 | 0.81 |
| TÜLU-1.1-70B | **0.79** | 0.78 | 0.75 | 0.77 | 0.45 | 0.86 | 0.64 | 0.84 | 0.86 |

Table 3: Results for TÜLU. **Bold** denotes the best performance in the output-only methods, while underline denote the best performance for the loss-based methods.

## 4.4   N-Gram Coverage Attack Generation Main Details

An important part of our pipeline is how much of the candidate member to use as the prefix. In our main experiments, we use 50% of the *words* from the candidate as the prefix. We also limit the generation length to be to the number of tokens removed, ensuring our input + output token budget is always $O(n)$, if the original text is length $n$. For BookMIA, the final results are reported with 100 generations per candidate; for everything else, we report with 50 generations due to computational constraints. See Appendix §C for more details.

## 4.5   Evaluation

Following prior work (Shi et al., 2023; Duan et al., 2024), we evaluate attack effectiveness using the area under the ROC curve (AUROC), rather than classification accuracy at a fixed threshold. AUROC provides a threshold-independent measure of how well an attack can distinguish between member and non-members – higher values indicate stronger attacks.

## 4.6   Main Results

Our comprehensive set of experiments shown in Table 1 – Table 3 demonstrate that N-Gram Coverage Attack is consistently effective across datasets, beating other black-box baselines for closed-models, and even performing close to white-box baselines

**N-Gram Coverage Attack consistently outperforms black-box baselines** We find N-Gram Coverage Attack **performs better than or equal to the other black-box baseline**, DE-COP, in all datasets. The closest comparison is shown in Table 2 for BookMIA: on the GPT-3.5 models, both black-box attacks perform well, but our method has a strong improvement over DE-COP on GPT-3.5 Instruct. Everywhere else, DE-COP struggles significantly behind N-Gram Coverage Attack, with near random performance, especially on the open-weight models like LLaMA 1 and TÜLU in Table 3. We hypothesize that this large drop-off in performance is due to the naive method assumption that the target model is a faithful question-answering model, which may not apply to weaker models.   **Simple n-gram coverage metrics can perform comparably to white-box attacks** Surprisingly, we find that N-Gram Coverage Attack also performs comparatively – or even better – to white-box attacks across all datasets as well. Specifically, it performs on average 95% as

well as the white-box baselines on the LLaMA models with WikiMIA in Table 1, and 91% as well with WikiMIA-24. Furthermore, we find that on WikiMIA$_{2024}$ Hard, our method **outperforms white-box attacks on all models**. Further results on the Pythia and OLMo models corroborate these results and are in Appendix §A.

**Coverage and Creativity consistently outperform LCS** We find that in general, the three n-gram similarity metrics we propose for N-GRAM COVERAGE ATTACK perform well. However, we find that across datasets, coverage and creativity consistently outperform the longest common substring, except for one model in BookMIA. We hypothesize this is because the coverage and creativity metrics 1) explicitly account for multiple matches, unlike LCS and 2) normalize by length, to contextualize the match length. Creativity and coverage, perform nearly equally otherwise, with coverage performing better on WikiMIA and WikiMIA$_{hard}$ 2024, where there are fewer matches in general, and creativity working better for BookMIA and TULU, which have more positive span matches to disambiguate.

**N-GRAM COVERAGE ATTACK is more efficient than black-box baselines** We compare the computational requirements of N-GRAM COVERAGE ATTACK to the existing black-box baseline, DE-COP (Duarte et al., 2024). Let $x$ be a candidate member of length $n$. DE-COP first generates three paraphrases of $x$, with an input length of $\approx n$ tokens and an output length of $\approx 3n$ tokens. The next step, multiple-choice question-answering, requires 24 generations of input length $\approx 4n$ and output length 1. The final token budget is $\approx 97n$ input and $\approx 3n$ output or approximately $100n$ total per sequence. In addition, it requires access to a powerful paraphraser model like Claude (Anthropic, 2023) for the initial stage, which further limits accessibility and incurs even more cost. On the other hand, N-GRAM COVERAGE ATTACK is more flexible, enabling a cost-performance tradeoff. Specifically, if we use index $k$ to construct a prefix $x_{\leq k}$, we can constrain our generation to be $n - k$ tokens. With $d$ generations, the total token budget becomes $d \times n$. We also make **no use of external models**, only relying on the target model itself for generation. Empirically, tested on WikiMIA$_{2024}$ Hard with LLaMA models and $d = 50$, DE-COP is computationally expensive, taking on average $2.6\times$ longer than our method despite performing much worse.

**Fine-tuning Membership Inference is Effective** Table 3 shows the detailed results for TÜLU. Across all model variants, most attacks, including N-GRAM COVERAGE ATTACK, can effectively determine membership with high accuracy; the notable exclusion is DE-COP. We also find the TÜLU 1.1 models display more resilience to attack compared to their equal-sized TÜLU 1 counterparts. We also find that reference models perform much poorer.

## 4.7 Ablation

We conduct additional experiments using BookMIA and GPT-3.5-0125 to further explore the impact of different hyperparameters, with important scaling conclusions.

**N-GRAM COVERAGE ATTACK scales with the number of sequences** The top of Figure 2 shows how performance scales with different N-Gram overlap metrics from N-GRAM COVERAGE ATTACK as we increase the number of sequences generated. For all metrics, scaling the size of the generations increases the attack performance. Intuitively, as we sample more generations from the model, we obtain a increasingly accurate output distribution representative of the true model probabilities. We also observe a similar scaling trend in other datasets, highlighting the versatility of our method.

**Given a fixed token budget, requesting the model to regenerate the last 50% of the sequence is best for performance** The middle of Figure 2 shows N-GRAM COVERAGE ATTACK performance as different proportions of the candidate document are used as the prefix. This is with a *fixed token budget* (which we use in main experiments), where the model can generate only as many tokens as exist in the suffix. Across n-gram overlap metrics, the best performing proportion is consistently at 50%. While more context is in-general helpful, since we have a fixed budget, using too large of a prefix limits both the suffix size and the generation length, which may harm performance.

**Temperature near 1.0 is consistently the best** We find that a temperature near 1.0 is important for the performance of N-GRAM COVERAGE ATTACK across metric. Though it might be

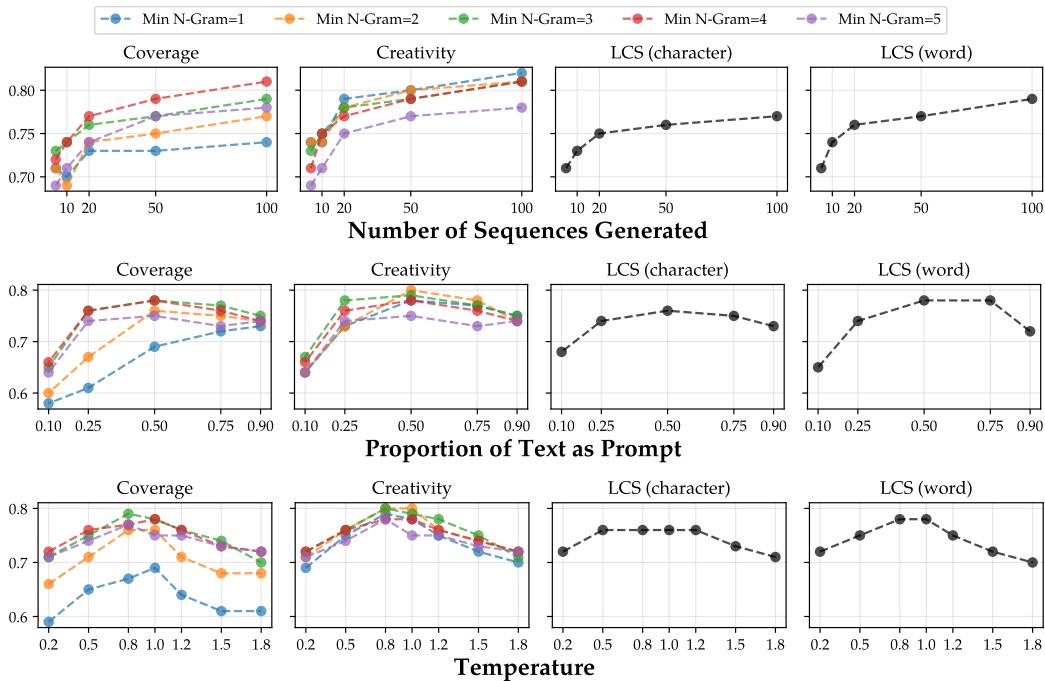

Figure 2: Scaling of N-GRAM COVERAGE ATTACK performance on BookMIA with different n-gram overlap metrics with GPT-3.5-0125 and `max` aggregation. We vary number of sequences generated (top), proportion of text used as prefix (mid.), and temperature (bot.).

expected that higher temperatures are in general more favorable, there is an intuitive trade-off between encouraging diversity to elicit harder-to-surface memorization and maintaining an accurate representation of the underlying distribution.

## 5 Conclusion

In this work, we introduce N-GRAM COVERAGE ATTACK, a membership inference attack that relies solely on text outputs from the target model, enabling attacks on completely black-box models. We demonstrate on a diverse set of benchmarks that N-GRAM COVERAGE ATTACK outperforms other black-box methods while also impressively achieving comparable or even better performance than state-of-the-art white-box attacks. We also find that our method is highly compute-efficient, scales well with increased repeated sampling, and its versatility allows us to investigate previously unstudied closed OpenAI models. Our findings reveals the vulnerability of language models, even in a fully black-box setting, underscoring the need for stronger privacy safeguards for large language models.

Overall, N-GRAM COVERAGE ATTACK provides a practical auditing tool for detecting problematic memorization, such as PII leakage or copyrighted content reproduction – critical concerns as models are trained on web-scale data of uncertain provenance. The method's efficiency and black-box nature make it valuable for monitoring deployed models and proactively identifying memorization risks. We hope this work encourages broader adoption of membership inference testing as part of responsible AI development.

## 6 Acknowledgments

We thank Johnny Wei and Robin Jia for their insightful suggestions on early versions of this work. We also appreciate helpful comments from Duygu Yaldiz and Yavuz Bakman. We also thank Jon May for his feedback and discussion on this work. Finally, we thank Mingma Sherpa for his constructive help throughout this project.

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

# A  Additional Experiments

We also conduct additional experiments for membership inference attacks on two datasets and families of open-access models. The same conclusions from the main text hold: N-GRAM COVERAGE ATTACK performs better than DE-COP and comparatively to the white-box baselines.

## A.1  Models

We use the following models:

**Pythia** (Biderman et al., 2023b) is a suite of decoder-only models from 70M to 12B parameters released by Eleuther AI. We use the 1.4B, 2.8B, 6.9B, and 12B models.

**OLMo** (Groeneveld et al., 2024) is a set of 1B and 7B models released by Ai2. We use the 07-024 checkpoints[5]. Finally, we also use the SFT and Instruction-tuned variants, which are further tuned to follow instructions and follow chat-style conversations respectively[6].

## A.2  Datasets

We use the following datasets:

**Dolma** (Soldaini et al., 2024) is a three-trillion token English corpus that was used to train OLMo (Groeneveld et al., 2024). It consists of text from diverse sources including books, scientific papers, code, and social media. We use the OLMo models as target models, as they were trained on Dolma. For out-members, we consider the Paloma evaluation suite (Magnusson et al., 2024), since passages that had an overlap with the Paloma evaluation suite were deliberately excluded from the Dolma pretraining corpus. We use Dolma-1.7, as it aligns with these OLMo checkpoints. Members are obtained from a random subset of Dolma[7], while non-members are obtained from the Paloma Dolma-v1.5 subset, which is dedpulicated against Dolma-1.7[8]. Our test set size is 1800 examples split evenly into members and non-members (sampled from the two datasets).

**The Pile** (Gao et al., 2020) is a massive corpus of English text designed for pretraining language models. Notably, it has been used to train the Pythia models (Biderman et al., 2023b) and LLama 1 (Touvron et al., 2023a), which become our target models. Pythia includes both training and test data, sampled from the same distribution independently, which becomes the gold members and non-members respectively. Previous studies (Duan et al., 2024) have found this to be a particularly challenging benchmark. Our test set size is a random subset of 1800 Pile members and non-members from Duan et al. (2024)[9], split evenly.

## A.3  Results

Our results are showm in Table 4 and Table 5. First, both tasks remain a challenging benchmark for all tasks, as performance is relatively low across the board, particularly with OLMo. N-GRAM COVERAGE ATTACK continues to outperform DE-COP even in this challenging setting, demonstrating again the strength of using only model generations and simple n-gram coverage metrics. N-GRAM COVERAGE ATTACK performs comparatively to the Pythia models, with scores near the loss and MinK baselines. On Dolma, N-GRAM COVERAGE ATTACK actually performs better in some cases – on OLMo-1B and OLMo-7B – than all white-box baselines, while performing close to the best-performing loss-based method, R-Loss, for the final two models.

---

[5]https://hf.co/allenai/OLMo-1B-0724-hf and https://hf.co/allenai/OLMo-7B-0724-hf

[6]https://hf.co/allenai/OLMo-7B-0724-Instruct-hf    and    https://hf.co/allenai/OLMo-7B-0724-SFT-hf

[7]https://hf.co/datasets/emozilla/dolma-v1_7-3B

[8]https://hf.co/datasets/allenai/paloma/viewer/dolma-v1_5

[9]https://hf.co/datasets/iamgroot42/mimir

Our conclusions here echo that in our main experiments: our method, particularly with coverage and creativity as similarity metrics, is effective across domains, and is comparable and **sometimes even better** than white-box attacks.

| Model | Output-Only Methods | | | | | Loss-Based Methods | | | |
|---|---|---|---|---|---|---|---|---|---|
| | Cov. | Cre. | LCS$_c$ | LCS$_w$ | D-C | Loss | R-Loss | zlib | MinK |
| Pythia 1.4B | 0.53 | 0.53 | 0.51 | 0.52 | 0.50 | 0.54 | 0.56 | 0.53 | 0.54 |
| Pythia 2.8B | 0.54 | 0.54 | 0.49 | 0.50 | 0.50 | 0.54 | 0.58 | 0.54 | 0.54 |
| Pythia 6.9B | 0.53 | 0.53 | 0.50 | 0.51 | 0.50 | 0.55 | 0.60 | 0.55 | 0.55 |
| Pythia 12B | 0.54 | 0.54 | 0.52 | 0.51 | 0.50 | 0.56 | 0.62 | 0.55 | 0.56 |

Table 4: Comparison of membership inference attack performance (AUROC) against the Pythia suite of models on the Pile. Across Pythia model scale, membership inference with the Pile remains challenging. **Bold** denotes the best performance in the output-only methods, while underline denotes the best performance for the loss-based methods.

| Model | Output-Only Methods | | | | | Loss-Based Methods | | | | Rand |
|---|---|---|---|---|---|---|---|---|---|---|
| | Cov. | Cre. | LCS$_c$ | LCS$_w$ | D-C | Loss | R-Loss | zlib | MinK | |
| OLMo-1B | 0.54 | 0.54 | 0.51 | 0.50 | 0.49 | 0.47 | - | 0.51 | 0.45 | |
| OLMo-7B | 0.54 | 0.54 | 0.54 | 0.51 | 0.5 | 0.47 | 0.53 | 0.51 | 0.46 | 0.49 |
| OLMo-7B-SFT | 0.52 | 0.52 | 0.53 | 0.51 | 0.5 | 0.47 | 0.53 | 0.51 | 0.46 | |
| OLMo-7B-Instruct | 0.52 | 0.52 | 0.52 | 0.51 | 0.5 | 0.47 | 0.52 | 0.51 | 0.46 | |

Table 5: Results for OLMo attacked with the DOLMa corpus. **Bold** denotes the best performance in the output-only methods, while underline denote the best performance for the loss-based methods.

# B  Experiments Details

We list further details of our experiments here, including more in-depth descriptions of loss-based baselines, hyperparemeters, and datasets.

## B.1  Implementation Details

We use HuggingFace (Wolf et al., 2019) to compute loss-based baselines. For all generation, including DE-COP and N-GRAM COVERAGE ATTACK, we use `vllm` for fast inference (Kwon et al., 2023).

## B.2  Baselines

We list further details of the baselines here, including hyperparemeters.

**Reference Loss**   For reference loss, we use smallest model of the same model family as the reference for the larger models. For example, for LLaMA 13B, 30B, and 65B, we use LLaMA 7B as the reference model. We do not run this baseline for the smallest model in the family.

**Min-K%** (Shi et al., 2023) measures the likelihood of the $k\%$ least-likely tokens (*outlier* tokens) in the given text under the target model, *i.e.*, $\text{Min-K}\%\,\text{PROB}(x) = \frac{1}{E}\sum_{x_i \in \text{Min-K}\%(x)} \log p\,(x_i \mid x_1, \ldots, x_{i-1})$, where $x$ is the input text and $E$ is the size of Min-K%$(x)$ set. A higher score indicates the model assigns unusually high likelihoods even to these rare tokens, suggesting potential memorization.

We run 6 variants, with $K$ set to 10% to 60% at 10% intervals, run these on our validation set and pick the best $K$ value before reporting the final test set.

**DE-COP** (**D-C**; Duarte et al. 2024) formulates membership inference as question-answering task. Given a text passage from a source of interest (*e.g.*, book), the method first synthesizes a set of QA pairs that ask which passage is a true excerpt from the source, juxtaposing the original passage against 3 synthetic paraphrases. The test statistics for membership inference is the accuracy of the target model on these QA tasks—intuitively, a model will mark high accuracy if the model has been trained on the source of interest. While the method works for black-box LLMs, it requires a strong paraphraser (*e.g.*, Claude (Anthropic, 2023)).

Following their implementation, we generate paraphrases using a temperature of 0.1 and with the prompts described in their paper; see Duarte et al. (2024). Since we do not have API access to Claude, we instead use a competitively capable GPT-4o model (OpenAI, 2024). For the multiple choice task, we use the same prompts for both closed and open source models that they list in their paper and on their Github repository[10].

### B.3   Models

We use the OpenAI API to access models. Not listed in the main experiments, we specifically use `gpt-4-turbo-2024-04-09` for GPT-4. For GPT-4o and 4o-mini, we use `gpt-4o-2024-04-09` and `gpt-4o-mini-0718` respectively.

### B.4   Datasets

**BookMIA** we select a random subset of 494 random book snippets from BookMIA for testing, due to extreme computational cost of the baseline and the cost of OpenAI models.

**WikiMIA** we select a random subset of 497 random article sumamries from WikiMIA testing, due to extreme computational cost of the baseline and the cost of OpenAI models.

## C   N-GRAM COVERAGE ATTACK Details

We list further details of our method N-GRAM COVERAGE ATTACK here.

### C.1   Method Parameters

For our main generations, we use nucleus sampling with a temperature of 1.0 and `top_p=0.95`. As mentioned above, we also constrain the max length of our generations to be the exact length (in tokens) of the suffix. For our aggregation metric, we use max in all experiments, as we find it to work the best in our small validation sets.

### C.2   Prompts

We list the final prompts which we use for our datasets in Table 6. For TULU, we do not use a prompt; we set the prefix to be the first user turn in chat, and allow the model to generate the full completion without any part of the ground truth response as context.

## D   Dataset Construction

We detail our steps to construct our additional datasets released in this work.

### D.1   WikiMIA$_{2024}$ Hard

We use the Wikimedia API to scrape random articles from Wikipedia; we filter out stubs, lists, and disambiguous pages. Next, we check to see 1) the page existed Dec 31, 2016 and 2) if there exists an edit in 2024 or later; if not, we discard. Next, we obtain the summaries

---

[10]https://github.com/LeiLiLab/DE-COP

---

**Prompt Templates for N-GRAM COVERAGE ATTACK**

**BookMIA**

> You will be shown a series of passages from famous literary works. After these examples,
> you will receive a prefix from another passage and be asked to complete it based on
> the text of a famous work. Provide only the continuation for the last given prefix
> without any extra commentary, formatting, or additional text.
>
> Complete the prefix: {prefix}

**WikiMIA/WikiMIA-24/WikiMIA$_{2024}$ Hard**

> Continue the generation as closely to verbatim as possible.
>
> {prefix}

**Pile and OLMo**

> Continue the text: {prefix}

---

Table 6: Prompts used for N-GRAM COVERAGE ATTACK across tasks. The {prefix} place-holder indicates where the input text is inserted.

of the page and check if 1) both the old and new version are at least 25 words long 2) The Levenshtein Edit Distance is above 0.5 (to ensure that there are sufficient differences and 3) The texts are no more than 20% different in their lengths. Following Shi et al. (2023), we keep only the first 256 words of the Wikipedia summary.

We identify and scrape 27000 Wikipedia pages which match the first criteria (existing in 2016, and having a valid edit) in approximately 3 hours. After deduplication and filtering for length and edit distance, we are left with 1040 instances, of which we randomly sample 1000. Overall, our final dataset consists of 2000 items, split evenly between members and non-members[11].

## D.2 TÜLU Dataset

TÜLU (Wang et al., 2023) is a collection of instruction-tuning datasets. We construct an MIA dataset by taking examples from the TÜLU Mix and examples from the datasets which were tested but not included; the full list is enumerated in Wang et al. (2023) and in Table 8. We use only the first-turn of these datasets.

We first attempt to randomly sample from both sets to create the dataset. However, the lengths are not very similar so perform binned sampling to ensure they are more even in length. First, we discard the bottom 5% shortest and top 5% longest sequences in both members and non-members to get rid of extreme responses. Next, we set $k = 10$ bins, evenly-space them, and sample from each dataset evenly in each bin to ensure that our datasets lengths are similar, and avoid spurious length correlations.

The statistics before and after pruning are shown in Table 7. Overall, our test set composition 924 members (from TÜLU), and 928 non-members from the other instruction datasets. Exact splits from each dataset is shown in Table 8

---

[11]We explore only Wikipedia in this case, but we could also construct a similar bookMIA 2024 set

| Length Type | Original | | After Sampling | |
|---|---|---|---|---|
| | **Member** | **Nonmember** | **Member** | **Nonmember** |
| User Length | 39.6 | 29.5 | 34.2 | 32.1 |
| Response Length | 27.9 | 25.2 | 26.5 | 25.1 |
| Total Length | 67.5 | 54.7 | 60.8 | 57.3 |

Table 7: Length Statistics Before and After Sampling for More Length Matching

| **Member** | | **Nonmember** | |
|---|---|---|---|
| **Category** | **Count** | **Category** | **Count** |
| GPT-4 Alpaca | 133 | Baize | 197 |
| OASST1 | 133 | Self Instruct | 201 |
| Dolly | 133 | Stanford Alpaca | 201 |
| Code Alpaca | 133 | Unnatural Instructions | 162 |
| ShareGPT | 133 | Super NI | 163 |
| Flan V2 | 133 | | |
| CoT | 126 | | |

Table 8: Member and Nonmember Dataset Representation

