# OpenReview forum: "The Surprising Effectiveness of Membership Inference with Simple N-Gram Coverage"
_colmweb.org/COLM/2025/Conference — COLM 2025_

### Official Review · Reviewer_zvq9 · 2025-05-08

**Rating:** 9
**Confidence:** 4
**Ethics Flag:** 1

**Summary:**

This paper proposes a membership inference attack method for large language models (LLM). The idea is to probe the LLM with prefix strings from a dataset and see if the suffix is generated. The advantage of the method is its simplicity. The authors show that it is a promising method on some proprietary and open-source models, though I do have some suggestions for improving the experiment and discussions:

1. It is nice that both proprietary and open-source models are used in experiments. However, we do not really know what is used in GPT-3.5/GPT-4, so as a matter of science there is no ground truth for your membership inference attack on those models. In fact, even for the various LLaMa variants, do you know for sure that the data samples you presume are used in training really did occur in training (it wasn't filtered)? I think it is fine to presume your new data is not in training, but the old ones are harder to say with certainty. There are several ways to address this:
- Be more clear about this as a caveat in Section 4.2
- Report both false positive and false negative results (or something similar that that breaks apart positive and negative samples, if you don't want to have a threshold) for at least one experiment.
- Provide more results on Tulu (or Olmo?) where it is more certain what was in training. Table 2 for example only shows GPT-3, so I don't know whether to fully trust the numbers.

2. More discussion on efficiency is warranted. The discussion in Section 4.6 says that the proposed method cost nxk tokens while previous work costs 100n, but is that difference important in practice? If I were a prosecutor representing book publishers who is using membership inference attack to decide whether to sue a LLM provider on copyright infringement, I may have a large budget. Further, I only need to find membership on just one book to provide evidence, i.e. I do not need multiple books as evidence to create a case. So I may be able to justify spending $1000 (I made up this number) on API calls since a lawsuit may be worth millions of dollars. So how exactly is your method "efficient" under this broader scheme?
- I would suggest adding a table that investigates efficiency of your method in more detail. It could be something like the top figures in Fig 2, but more in-depth. For example, vary d and n and perhaps k and give recommendation on what is the recommended setting. It may also be interesting to try on sequences that have relatively frequent words vs those that don't.

3. This is a minor point, but the discussion of "white-box methods" and "black-box methods" could potentially confuse the reader because these terms were not defined.
- Do you say clearly "white-box" means that likelihood information is used, and that it is not white-box in the sense that internal parameters are used. Some API may provide a score for the generation so the distinction is important.
- Can you do "white-box methods" on "black-box models"? It's not clear if methods and models are tied.
- Many of the results have statements like "N-gram coverage attack is more efficient than black-box methods" where methods is plural form but really it is just one method. It is better to just say your method is better DE-COP rather than making a seemingly general claim of improvement over ALL black-box methods.

4. It may be worthwhile citing the following paper. They show that using a group of sentences (rather than a single sentence) with ngram features can lead to improvements in the membership inference attack. The method is different and their setting focuses on machine translation sequence-to-sequence models, but their result corroborates well with your idea of using multiple strings and ngram overlaps.

Hisamoto (2020). Membership Inference Attacks on Sequence-to-Sequence Models: Is My Data In Your Machine Translation System? https://direct.mit.edu/tacl/article/doi/10.1162/tacl_a_00299/43536/Membership-Inference-Attacks-on-Sequence-to

**Reasons To Accept:**

1. Membership inference for LLMs is an important problem considering the advances in the field and ongoing lawsuits regarding copyright, etc.
2. The paper proposes a simple but promising solution. The experiments and comparisons are insightful.

**Reasons To Reject:**

1. Membership inference is a problem that potentially has important ramifications: for example, a prosecutor may use this to decide whether to sue LLM providers for copyright infringement. Because of this, I think there needs to be more rigorous and nuanced discussion of results (beyond simply stating that the method is "effective."). That said, I gave some suggestions (see "Summary") and I think it is not too hard to address them in the revisions.

---

> ### Author Response · Authors · 2025-06-03
> **Response to Reviewer zvq9 (Part 1)**
>
> Thank you for your review - we truly appreciate your detailed feedback and thoughtful suggestions! We are grateful for your recognition that membership inference for LLMs is "an important problem" and that our approach is "simple but promising". We’re also glad that you found our experiments “insightful” and acknowledge the value of evaluating both proprietary and open-source models.
>
> We address all your questions and concerns below, and would be happy to discuss any remaining points during the discussion period.
>
> ---
>
> **Concern 1**: *It is nice that both proprietary and open-source models are used in experiments. However, we do not really know what is used in GPT-3.5/GPT-4, so as a matter of science there is no ground truth for your membership inference attack on those models. In fact, even for the various LLaMa variants, do you know for sure that the data samples you presume are used in training really did occur in training (it wasn't filtered)? I think it is fine to presume your new data is not in training, but the old ones are harder to say with certainty. There are several ways to address this:*
> * *Be more clear about this as a caveat in Section 4.2*
> * *Report both false positive and false negative results (or something similar that breaks apart positive and negative samples, if you don't want to have a threshold) for at least one experiment.*
> * *Provide more results on Tulu (or Olmo?) where it is more certain what was in training. Table 2 for example only shows GPT-3, so I don't know whether to fully trust the numbers.*
>
> Thank you for your appreciation of our evaluation on both proprietary and open-source models, and for this important observation about ground truth uncertainty. You raise a valid concern that we share with the broader membership inference community.
>
> We agree that evaluating membership inference attacks on closed-source models (like GPT-3.5/GPT-4) and even some open-source models (like LLaMA) introduces uncertainty, as the training data for these models is not publicly confirmed. Though the WikiMIA [1] paper justifies: *“For member data, we collected articles created before 2017 because many pretrained models, e.g., LLaMA, GPT-NeoX and OPT, were released after 2017 and incorporate Wikipedia dumps into their pretraining data.”*,  we agree that this is ultimately a silver-standard assumption, not a ground truth. Similarly, BookMIA [1] uses *presumed members* collected from [2], which employs name cloze queries to identify *likely book members* in OpenAI models. Ultimately, while these two datasets use assumptions on members, rather than guarantees, i.e. silver-standard members and gold non-members, they represent the best available approximations to benchmark many state-of-the-art models such as LLaMa and ChatGPT, given that model providers rarely disclose complete training datasets. **However, we think all of your suggestions would strengthen our paper in the next version.**
> * We will make the uncertainty of member data in WikiMIA and BookMIA more clear as a caveat in Section 4.2, when we introduce them. Specifically, we will explicitly acknowledge that member data for WikiMIA and BookMIA should be considered "high-confidence presumed members" (ie, silver rather than gold labels) rather than guaranteed members.
> * We will report performance broken down by member/non-member samples to provide transparency about our method's behavior on each class. **Specifically, we have computed TPR at low FPR results at different thresholds (0.01%, 0.1%, 0.5%, 1.0%, 5.0%) in response to Reviewer Yujw, which we will also report in the next version of the paper.** These results confirm that across domains, our method N-Gram Coverage Attack largely achieves strong precision in low-FPR regimes.
> * We agree that evaluating on more models with more clearly known training data would strengthen the work. For the next version of the paper, we will expand our evaluation to include Tulu-2 models [3] and OLMo-2 variants (1B - 32B) [4], where training data is publicly documented, to complement our results on Tulu [8] in the main text, and our results on the Pile [5] (via MIMIR [6]) and OLMo [7]  in the Appendix. This will strengthen our main claims.
>
> In summary, while we acknowledge the limitations in ground-truth labels for some benchmarks, the combination of our expanded evaluation, proposed clarifications, and additional experiments on models with documented training data will ensure our conclusions remain robust. We appreciate this suggestion and believe the resulting improvements will further strengthen the clarity and scientific rigor of our paper.

---

> > ### Author Response · Authors · 2025-06-03
> > **Response to Reviewer zvq9 (Part 2)**
> >
> > ---
> > **Concern 2** : *More discussion on efficiency is warranted. The discussion in Section 4.6 says that the proposed method cost nxk tokens while previous work costs 100n, but is that difference important in practice? If I were a prosecutor representing book publishers who is using membership inference attack to decide whether to sue a LLM provider on copyright infringement, I may have a large budget. Further, I only need to find membership on just one book to provide evidence, i.e. I do not need multiple books as evidence to create a case. So I may be able to justify spending $1000 (I made up this number) on API calls since a lawsuit may be worth millions of dollars. So how exactly is your method "efficient" under this broader scheme?*
> > * *I would suggest adding a table that investigates efficiency of your method in more detail. It could be something like the top figures in Fig 2, but more in-depth. For example, vary d and n and perhaps k and give recommendation on what is the recommended setting. It may also be interesting to try on sequences that have relatively frequent words vs those that don't.*
> >
> > Thank you for this thoughtful and constructive suggestion! You raise a great point about the context-dependent nature of "efficiency." We agree that for high-stakes applications like litigation, API costs may be negligible compared to potential outcomes.
> >
> > In light of this, we will clarify that our notion of efficiency is not intended to argue that other methods are infeasible in absolute terms, but rather to emphasize relative cost-effectiveness. Specifically, our method achieves better performance than the prior black-box attacks DE-COP while using significantly fewer tokens – a reduction that becomes meaningful when applied across many queries, targets, or models. For example, if the goal is not just to investigate one book, but to screen hundreds of documents, run audits across model versions, or conduct repeated testing as APIs evolve, token costs can quickly become the bottleneck.
> >
> > We will revise our paper to acknowledge that efficiency is not universally critical; for single high-stakes investigations, absolute costs may indeed be acceptable. Instead, we will emphasize how our cost reduction enables broader accessibility (e.g., to researchers or journalists) and scalability (e.g., for institutions conducting regular audits or detecting systemic data leaks).
> >
> > Finally, we agree it would strengthen the paper to contextualize efficiency more concretely. In the next version of the paper **we will add a comprehensive table showing performance versus cost tradeoffs for varying prefix lengths, sample sizes, and suffix lengths, with specific recommendations** for different scenarios ranging from single-target investigation to large-scale auditing. Following your excellent suggestion, we will also analyze how efficiency varies between sequences with frequent versus rare tokens sequences, as rarity may affect the number of samples needed for reliable detection.
> >
> > ---
> >
> > **Concern 3**: *This is a minor point, but the discussion of "white-box methods" and "black-box methods" could potentially confuse the reader because these terms were not defined.*
> > * *Do you say clearly "white-box" means that likelihood information is used, and that it is not white-box in the sense that internal parameters are used. Some API may provide a score for the generation so the distinction is important.*
> > * *Can you do "white-box methods" on "black-box models"? It's not clear if methods and models are tied.*
> > * *Many of the results have statements like "N-gram coverage attack is more efficient than black-box methods" where methods is plural form but really it is just one method. It is better to just say your method is better DE-COP rather than making a seemingly general claim of improvement over ALL black-box methods.*
> >
> > Thank you for raising this point. We appreciate you pointing out the ambiguity in our use of 'white-box' and 'black-box'. We agree this needs clarification, and we will add clear definitions in Section 2 (Background and Related Work), and mention this again when we introduce our method in Section 3.
> >
> > We define **black-box methods** as attacks that rely solely on text outputs from the model—i.e., access to completions without probabilities or internal state. We define **white-box methods** as attacks requiring additional internal signals, including parameters, token-level log probabilities, or gradients. While some APIs may expose token probabilities, we still classify methods that *leverage* these internal scoring signals as *white-box*, since they rely on privileged access beyond text generation. This allows us to draw a clear line between surface-level outputs (black-box) and internal signals (white-box), even if both are accessed through APIs.

---

> > > ### Author Response · Authors · 2025-06-03
> > > **Response to Reviewer zvq9 (Part 3)**
> > >
> > > We agree with your point that “white-box” can refer to different types of access – e.g., gradients vs. log-likelihoods – and thus exists on a spectrum. In our paper, we treat all methods requiring any internal scoring signals as white-box without further sub-classification for clarity. We will also clarify this explicitly in the next version of our paper in Section 2.
> > >
> > > Regarding whether methods and models are tied, we clarify that methods are defined by what information they *require*, and models by what information they *expose*. A white-box method (e.g., one requiring token likelihoods) can only be applied to models that expose the necessary internal signals (ie, a white-box model with respect to token likelihoods). Therefore, such a method cannot be used on a model that only provides completions (aka, a black-box model). Conversely, black-box methods are broadly applicable across all models, as they require only surface-level access to generated outputs. We will clarify this distinction to avoid ambiguity between method and models access levels.
> > >
> > > Finally, thank you for noting the phrasing regarding baselines. We will revise our phrasing throughout the text to avoid implying that our method outperforms *all* black-box methods, and instead explicitly name the baseline (e.g., “our method is more efficient than DE-COP”).
> > >
> > > ---
> > >
> > > **Concern 4**: *It may be worthwhile citing the following paper. They show that using a group of sentences (rather than a single sentence) with ngram features can lead to improvements in the membership inference attack. The method is different and their setting focuses on machine translation sequence-to-sequence models, but their result corroborates well with your idea of using multiple strings and ngram overlaps.*
> > > * *Hisamoto (2020). Membership Inference Attacks on Sequence-to-Sequence Models: Is My Data In Your Machine Translation System? https://direct.mit.edu/tacl/article/doi/10.1162/tacl_a_00299/43536/Membership-Inference-Attacks-on-Sequence-to*
> > >
> > > Thank you for this excellent suggestion! We appreciate you pointing out this highly relevant work. While they focus on machine translation, their demonstration that n-gram features are effective for membership inference in sequence-to-sequence models provides useful corroboration for our approach.
> > >
> > > We will add this to our related work section**, acknowledging their contribution to demonstrating that n-gram-based features can be effective signals for membership inference, which supports exploring such approaches in other settings like ours. Thank you for this helpful reference!
> > >
> > > ---
> > >
> > > **Concern 5:** *Membership inference is a problem that potentially has important ramifications: for example, a prosecutor may use this to decide whether to sue LLM providers for copyright infringement. Because of this, I think there needs to be more rigorous and nuanced discussion of results (beyond simply stating that the method is "effective.").*
> > >
> > > Thank you for highlighting this important point about the societal implications of membership inference attacks. We agree that given the potential legal and ethical ramifications, our discussion would benefit more nuance and rigor.
> > >
> > > **We will revise the paper to more clearly articulate what “effective” means in our context, grounding our claims in specific quantitative results** (e.g., AUROC, TPR at low FPR) and discussing their real-world implications. We will also explicitly scenarios where high precision is required, such as legal decision-making, and explain the tradeoffs involved in applying our method under such constraints. We also commit to qualifying our claims more rigorously and will avoid any language that could be interpreted as overstating our findings.
> > >
> > > **Additionally, we will add a dedicated "Limitations and Societal Impact" section** that discusses the evidentiary limitations of membership inference attacks. Specifically, we will address: (1) the probabilistic nature of our method and what confidence scores imply in practice, (2) the fact that membership inference is inherently circumstantial and may require corroborating evidence in legal or forensic settings, and (3) the potential for both beneficial applications (e.g., transparency, auditing) and harmful misuse.
> > >
> > > We recognize that our work could impact important debates around AI training data and copyright. By providing more rigorous analysis and careful framing, we aim to contribute responsibly to this discussion. Thank you for pushing us to address these crucial considerations.

---

> > > > ### Author Response · Authors · 2025-06-03
> > > > **Response to Reviewer zvq9 (Part 4)**
> > > >
> > > > # References
> > > > [1] Shi, W., Ajith, A., Xia, M., Huang, Y., Liu, D., Blevins, T., Chen, D., & Zettlemoyer, L.S. (2023). Detecting Pretraining Data from Large Language Models. ArXiv, abs/2310.16789.
> > > >
> > > > [2] Chang, K.K., Cramer, M.H., Soni, S., & Bamman, D. (2023). Speak, Memory: An Archaeology of Books Known to ChatGPT/GPT-4. ArXiv, abs/2305.00118.
> > > >
> > > > [3] Ivison, H., Wang, Y., Pyatkin, V., Lambert, N., Peters, M.E., Dasigi, P., Jang, J., Wadden, D., Smith, N.A., Beltagy, I., & Hajishirzi, H. (2023). Camels in a Changing Climate: Enhancing LM Adaptation with Tulu 2. ArXiv, abs/2311.10702.
> > > >
> > > > [4] OLMo, T., Walsh, P., Soldaini, L., Groeneveld, D., Lo, K., Arora, S., Bhagia, A., Gu, Y., Huang, S., Jordan, M., Lambert, N., Schwenk, D., Tafjord, O., Anderson, T., Atkinson, D., Brahman, F., Clark, C., Dasigi, P., Dziri, N., Guerquin, M., Ivison, H., Koh, P.W., Liu, J., Malik, S., Merrill, W., Miranda, L.V., Morrison, J.D., Murray, T.C., Nam, C., Pyatkin, V., Rangapur, A., Schmitz, M., Skjonsberg, S., Wadden, D., Wilhelm, C., Wilson, M., Zettlemoyer, L.S., Farhadi, A., Smith, N.A., & Hajishirzi, H. (2024). 2 OLMo 2 Furious. ArXiv, abs/2501.00656.
> > > >
> > > > [5] Gao, L., Biderman, S., Black, S., Golding, L., Hoppe, T., Foster, C., Phang, J., He, H., Thite, A., Nabeshima, N., Presser, S., & Leahy, C. (2020). The Pile: An 800GB Dataset of Diverse Text for Language Modeling. ArXiv, abs/2101.00027.
> > > >
> > > > [6] Duan, M., Suri, A., Mireshghallah, N., Min, S., Shi, W., Zettlemoyer, L.S., Tsvetkov, Y., Choi, Y., Evans, D., & Hajishirzi, H. (2024). Do Membership Inference Attacks Work on Large Language Models? ArXiv, abs/2402.07841.
> > > >
> > > > [7] Groeneveld, D., Beltagy, I., Walsh, P., Bhagia, A., Kinney, R., Tafjord, O., Jha, A., Ivison, H., Magnusson, I., Wang, Y., Arora, S., Atkinson, D., Authur, R., Chandu, K.R., Cohan, A., Dumas, J., Elazar, Y., Gu, Y., Hessel, J., Khot, T., Merrill, W., Morrison, J.D., Muennighoff, N., Naik, A., Nam, C., Peters, M.E., Pyatkin, V., Ravichander, A., Schwenk, D., Shah, S., Smith, W., Strubell, E., Subramani, N., Wortsman, M., Dasigi, P., Lambert, N., Richardson, K., Zettlemoyer, L.S., Dodge, J., Lo, K., Soldaini, L., Smith, N.A., & Hajishirzi, H. (2024). OLMo: Accelerating the Science of Language Models. Annual Meeting of the Association for Computational Linguistics.
> > > >
> > > > [8] Wang, Y., Ivison, H., Dasigi, P., Hessel, J., Khot, T., Chandu, K.R., Wadden, D., MacMillan, K., Smith, N.A., Beltagy, I., & Hajishirzi, H. (2023). How Far Can Camels Go? Exploring the State of Instruction Tuning on Open Resources. ArXiv, abs/2306.04751.

---

> > > > > ### Comment · Reviewer_zvq9 · 2025-06-05
> > > > >
> > > > > Thanks for the detailed response! I think the paper is already very nice, and with the additional revisions stated it will become even stronger.  Good luck!

---

> > > > > > ### Author Response · Authors · 2025-06-10
> > > > > >
> > > > > > Thank you for your thoughtful feedback! We truly appreciate your support in strengthening the paper.

---

### Official Review · Reviewer_avj5 · 2025-05-13

**Rating:** 4
**Confidence:** 5
**Ethics Flag:** 1

**Summary:**

This paper studies an interesting problem, but there are flaws in the contribution, method, related works, etc.

**Questions To Authors:**

Additional comment: A random-guess baseline is not necessary; please consider removing it from the results. It is obvious that the AUC of a random guess is 0.5.

**Reasons To Accept:**

(1) The focus on the full black-box MIAs requires fewer assumptions than previous methods.
(2) This work is the first to explore the effectiveness of N-GRAM attack on post-training data, a very interesting perspective.

**Reasons To Reject:**

(1) One major contribution of this paper is the collection of a new benchmark, WIKIMIA-2024, which, however, is almost identical to the benchmark WIKIMIA-24 released in [1]. Both benchmarks are based on WIKIMIA but adapted to up-to-date LLMs by setting a new cut-off date. The paper should discuss in detail the essential differences between these two benchmarks, if they exist. Additionally, evaluating the proposed method and baselines on WIKIMIA-24 is also necessary.
(2) What is the meaning of edit distance (Line 210)? Why should we ensure a sufficient distance between the member and non-member? The necessity of this operation is not justified.
(3) The related works/preliminaries can benefit from a thorough revision, especially since the discussion of MIAs and Memorization is not comprehensive and up-to-date enough. Please include a detailed discussion of more up-to-date works [1-5].
(4) As established by Carlini et al. [6], membership inference attacks should be evaluated by computing their true-positive rate at low (e.g., ≤ 0.1%) false-positive rates, rather than using average-case metrics like AUC. Such evaluation metrics have become the de facto standard in evaluating membership inference attacks and are used by many existing works [5] [7] [8]. I suggest using two metrics: the Full Log-scale Receiver Operating Characteristic (ROC) Curve to highlight low false-positive rates, and the TPR at a low FPR, which measures attack performance at specific FPRs (e.g., 0.1%, 0.01%).


[1]	Fu, Wenjie, et al. "MIA-Tuner: Adapting Large Language Models as Pre-training Text Detector." arXiv e-prints (2024): arXiv-2408.
[2]	Garg, Isha, et al. “Memorization through the Lens of Curvature of Loss Function around Samples.” Proceedings of the 41st International Conference on Machine Learning, PMLR, 2024.
[3]	Ravikumar, Deepak, et al. “Unveiling Privacy, Memorization, and Input Curvature Links.” Proceedings of the 41st International Conference on Machine Learning, PMLR, 2024.
[4]	Zhang, Jingyang, et al. Min-K%++: Improved Baseline for Detecting Pre-Training Data from Large Language Models. arXiv:2404.02936, arXiv, 23 May 2024.
[5]	Fu, Wenjie, et al. "Membership Inference Attacks against Fine-tuned Large Language Models via Self-prompt Calibration." The Thirty-eighth Annual Conference on Neural Information Processing Systems. 2024.
[6]	Carlini, Nicholas, et al. "Membership inference attacks from first principles." 2022 IEEE Symposium on Security and Privacy (2022).
[7]	Bertran, Martin, et al. "Scalable membership inference attacks via quantile regression." Advances in Neural Information Processing Systems (2023).
[8]	Wen, Yuxin, et al. "Canary in a Coalmine: Better Membership Inference with Ensembled Adversarial Queries." International Conference on L

---

> ### Author Response · Authors · 2025-06-03
> **Response to Reviewer avj5 (Part 1)**
>
> Thank you for your review!  We're glad you found the problem interesting and appreciated our focus on “full black-box MIAs,” which “require fewer assumptions than previous methods.” We agree that this flexibility makes our approach especially well-suited for evaluating a wide range of models, including fully closed-access systems. We're also encouraged that you recognized our work as the first to explore the effectiveness of N-Gram attacks on post-training data, and appreciated it as “a very interesting perspective.”
>
> We address all your questions and concerns below, and would be happy to discuss any remaining points during the discussion period.
>
> ---
>
> **Concern 1**: *Additional comment: A random-guess baseline is not necessary; please consider removing it from the results. It is obvious that the AUC of a random guess is 0.5.*
>
> Thank you for this suggestion! We will remove it from the results tables in the next version of the paper.
>
> ---
>
> **Concern 2**: *The related works/preliminaries can benefit from a thorough revision, especially since the discussion of MIAs and Memorization is not comprehensive and up-to-date enough. Please include a detailed discussion of more up-to-date works [1-5].*
>
> Thank you for these valuable references. We appreciate you pointing out recent works that would strengthen our literature review. **We will expand our related work section to include detailed discussion of [1-5] in the revised version.**
>
> Specifically, here is how we plan to incorporate these works in our related work section (*note: we cite [1] as Fu et al. 2025 following its AAAI publication*):
> * We will insert discussion of [1] into line 81 (Background and Related Work; Membership Inference): *Fu et al. (2025) demonstrate that fine-tuning large language models can enable effective membership inference detection; however, this approach assumes that the model provider permits fine-tuning, besides requiring supervision.*
> * We will add discussion of [2] and [3] after line 94 (Background and Related Work; Memorization): *There have also been several efforts that aim to uncover memorization evidence, assuming access to the model’s prediction loss over a sequence (Garg et al. 2024; Ravikumar et al. 2024). In contrast, our work is based on the fully blackbox setting, where we only assume API-level access to the model.*
> * Although we cite [4] already on line 80 (Background and Related Work; Membership Inference), we agree it deserves more detailed discussion. We will write the following: *Zhang et al. (2024b) later extend upon Min-K by leveraging statistics from the entire vocabulary distribution to normalize token probabilities, showing improved results; the key intuition is changing the membership signal from absolute to relative token probabilities.*
> * Since [5] is an attack based on fine-tuning the reference model itself, we will add it as a citation to line 76 (Background and Related Work; Membership Inference), where we extensively list out other reference-model based attacks: *While Carlini et al. (2020) additionally use a reference model – a language model with less memorization – to remove the effect of intrinsic sequence difficulty from the observed loss, this technique has found widespread use (Mattern et al., 2023; Mireshghallah et al., 2022; Ye et al., 2021; **Fu et al., 2024**). However, it is difficult to ascertain whether the reference model itself has memorized the sequence…*

---

> > ### Author Response · Authors · 2025-06-03
> > **Response to Reviewer avj5 (Part 2)**
> >
> > ---
> > **Concern 3**: *(4) As established by Carlini et al. [6], membership inference attacks should be evaluated by computing their true-positive rate at low (e.g., ≤ 0.1%) false-positive rates, rather than using average-case metrics like AUC. Such evaluation metrics have become the de facto standard in evaluating membership inference attacks and are used by many existing works [5] [7] [8]. I suggest using two metrics: the Full Log-scale Receiver Operating Characteristic (ROC) Curve to highlight low false-positive rates, and the TPR at a low FPR, which measures attack performance at specific FPRs (e.g., 0.1%, 0.01%).*
> >
> > Thank you for this suggestion. We agree that TPR at low FPR is another important metric for evaluating membership inference attacks. Following your recommendation and previous work [1,4,5,8,10,12-20], **we have expanded our evaluation across all datasets to include TPR at multiple low FPR thresholds (0.1%, 0.5%, 1.0%, 5.0%)** – standard thresholds across prior work [1,4,5,7-8,10,12-25] – alongside AUROC. We also agree that log-scale ROC curve visualizations are valuable to further contextualize method performances, and **will add these in the next version of the paper.**
> >
> > We view AUROC and TPR at low FPR as complementary metrics: AUROC captures overall discriminative power, while TPR at low FPR measures performance in practical settings where false positives are especially undesirable. **Our results demonstrate strong performance on both metrics, validating our method's effectiveness across different evaluation paradigms**.
> >
> > Please see **Discussion of New Results with TPR @ low FPR thresholds** below for detailed analyses and Section **Updated Experiment Results** below for the specific tables.
> >
> >
> > ## **Discussion of New Results with TPR @ low FPR thresholds**
> >
> > **Our updated results demonstrate strong performance on both metrics**. First, as shown in the original results (reproduced in the updated tables for completeness), our method maintains its superiority in AUROC over DE-COP across all benchmarks while remaining competitive with white-box attacks. More importantly, **the newly-added TPR at low FPR results confirm our method's strong performance: consistently outperforming black-box baselines while achieving competitive performance to white-box methods across diverse operating points**.
> >
> > We analyze the TPR at low FPR results more in depth. On BookMIA, our method achieves exceptional performance, convincingly outperforming the black-box baseline DE-COP at all FPR thresholds and on all models. Even at the stringent 0.1% FPR threshold, we achieve high TPR values (e.g., a minimum of 0.39 across models), demonstrating strong discriminative power in low false-positive regimes.
> >
> > For the Tulu dataset, we consistently surpass DE-COP which struggles at all thresholds. Notably, our method remains competitive with white-box approaches and even exceeds them at moderate FPR thresholds (0.5%, 1.0%) for the 70B Tulu model, suggesting that our approach can sometimes even exceed white-box in certain scenarios.
> >
> > The WikiMIA_2024 Hard results mirror the strong Tulu findings, with our method outperforming DE-COP across all thresholds and models except one, while showing competitive performance against white-box methods. For example, at 1.0% FPR on LLaMA 65B, we slightly outperform all white-box attacks.
> >
> > On WikiMIA, our method and DE-COP exhibit a clear tradeoff. At 1.0% FPR, DE-COP performs slightly better overall, outperforming our method on 4 of 7 models, though the margins are often small (e.g., 0.01). However, at 5.0% FPR, which slightly relaxes the threshold but remains realistic for many applications [4,10, 20, 23-25], our method surpasses DE-COP on 6 out of 7 models. This aligns with our higher AUROC and suggests that for this dataset, our method provides more consistent signal at moderate FPRs, while DE-COP peaks earlier but plateaus. This tradeoff pattern – where different methods excel at different operating points – underscores the importance of evaluating across multiple FPR thresholds rather than relying on a single metric.
> >
> > Finally, across all benchmarks, **our method exhibits strong performance and consistent improvement as FPR thresholds increase, with steady TPR growth that contrasts with DE-COP's tendency to stagnate**, particularly on harder benchmarks. Furthermore, our method typically achieves meaningful TPR values even at 0.1% FPR, indicating strong early detection capabilities. These results validate that our approach not only excels in average-case performance (AUROC) but also delivers practical value in low false-positive scenarios where precision is important.

---

> > > ### Author Response · Authors · 2025-06-03
> > > **Response to Reviewer avj5 (Part 3)**
> > >
> > > (continued from Response to Reviewer avj5 (Part 2)
> > >
> > > **Overall, our additional results with TPR at low FPR across four diverse datasets confirm our strong AUROC results**, showing that our method is **consistently effective** across model families, data domains, and evaluation metrics. This robustness combined with our method's simplicity and efficiency, validates our design choices and establishes N-Gram Coverage Attack as a practical and effective black-box membership inference attack.
> > >
> > > ---
> > >
> > > **Concern 4**: *(1) One major contribution of this paper is the collection of a new benchmark, WIKIMIA-2024, which, however, is almost identical to the benchmark WIKIMIA-24 released in [1]. Both benchmarks are based on WIKIMIA but adapted to up-to-date LLMs by setting a new cut-off date. The paper should discuss in detail the essential differences between these two benchmarks, if they exist. Additionally, evaluating the proposed method and baselines on WIKIMIA-24 is also necessary.*
> > >
> > > *For clarity, we denote the benchmark we release as **WikiMIA_2024 Hard**, and the benchmark from [1] as **WikiMIA-24**.*
> > >
> > > **Clarification of Contributions**
> > >
> > > We would like to gently clarify that while the WikiMIA_2024 Hard dataset is one of the contributions of our paper, **our primary contributions center on: (1) the N-Gram Coverage Attack** – a simple, scalable black-box method that often matches white-box performance, and **(2) our systematic analysis**, revealing how design choices (aggregation strategies, sampling methods, similarity metrics) substantially impact performance.
> > >
> > > WikiMIA_2024 Hard supports these contributions by providing a cleaner evaluation setting with reduced distribution shift artifacts. We view it as an important but auxiliary contribution that enables more rigorous evaluation of our method and findings. That said, we recognize the importance of distinguishing WikiMIA_2024 Hard from WikiMIA-24, and provide detailed comparison below.
> > >
> > > **Differences between WikiMIA_2024 Hard and WikiMIA-24 datasets**
> > >
> > > Thank you for raising this important distinction between our WikiMIA_2024 Hard dataset and the existing WikiMIA-24 dataset [1]. While both benchmarks apply a similar strategy of updating the cutoff date to adapt to more recent LLMs, their construction methodology differs significantly.
> > >
> > > The original WikiMIA dataset [10] is collected by using different Wikipedia articles for members (pre-2017 event pages) and non-members (post-2023 event pages). While useful for initial evaluation, [9] demonstrates that such temporal splits can introduce distribution shifts, where models might leverage spurious features rather than detecting actual memorization.
> > >
> > > **WikiMIA-24 follows the original WikiMIA [10] collection process nearly identically, only updating the cutoff dates**. The authors state they "follow a similar pipeline of WIKIMIA to collect a more up-to-date benchmark by moving forward the cutoff date to March 1, 2024" and "follow the same setting of WIKIMIA" for member data (Section 5.1, Benchmark Datasets Construction). Because WikiMIA-24 only updates the cutoffs, making no other modifications to the data collection procedure of WikiMIA [10], it **likely inherits the same distribution shift vulnerabilities identified in [9]**.
> > >
> > > **Our WikiMIA_2024 Hard, in contrast, employs a fundamentally different construction approach than in WikiMIA [10] and WikiMIA-24 [1] to specifically address this distribution shift**. Unlike these benchmarks which use completely different articles for members vs non-members, WikiMIA_2024 Hard leverages different versions of the same Wikipedia article (pre-2017 and post-2024) to create member/non-member pairs. This ensures that members and non-members are topically aligned and differ only due to the natural evolution of the same entry, rather than being drawn from distinct temporal event sets. By construction, this approach creates **a cleaner, lower-shift benchmark** that reduces spurious temporal signals. We validate this empirically by showing consistently lower performance across all methods and attacks on WikiMIA_2024 Hard compared to WikiMIA and WikiMIA-24. See Section **Additional Experiments on WikiMIA-24* directly below for detailed analysis. These steps and justifications are described further in Section 4.2 (Lines 200–211) of our paper. **We will also clarify these distinctions between WikiMIA-2024 and WikiMIA-24 explicitly in the next version of the paper.**

---

> > > > ### Author Response · Authors · 2025-06-03
> > > > **Response to Reviewer avj5 (Part 4)**
> > > >
> > > > (continued from Response to Reviewer avj5 (Part 3)
> > > >
> > > > **Additional Experiments on WikiMIA-24**
> > > >
> > > > Following your suggestion, **we run additional experiments on WikiMIA-24 [1]**. We evaluate on the same set of models we test on WikiMIA_2024 Hard. Our result tables are shown in Section **Updated Experiment Results**. We highlight two key findings: (1) our method generalizes well to WikiMIA-24 and (2) WikiMIA_2024 Hard is significantly more challenging, validating its utility as a rigorous evaluation benchmark, and indicating the reduction of spurious temporal signals which exist in WikiMIA-24 and WikiMIA.
> > > >
> > > > First, our method's strong performance generalizes to WikiMIA-24. We maintain clear superiority over DE-COP, consistently outperforming it across models and datasets, and remain competitive with white-box attacks on both AUROC and TPR at low FPR metrics. This demonstrates that our method’s strong performance is not benchmark-specific, but generalizes to datasets constructed using different methodologies.
> > > >
> > > > Second, **the results validate that the reduced distribution shift of WikiMIA_2024 Hard removes exploitable artifacts and creates a stricter test of true memorization**. WikiMIA_2024 Hard is substantially more challenging than WikiMIA-24: the performance gap is large across all methods and metrics. For example, on WikiMIA-24, across all models, the best AUROC score reaches 0.84 (on GPT-4), while the maximum on WikiMIA_2024 Hard is only 0.64 (also on GPT-4). Furthermore, using TPR @ 1% FPR, the best performing attack on any OpenAI model is 0.23 for WikiMIA-24, but only 0.08 on WikiMIA_2024 hard. Finally, on WikiMIA-24, a significant number of white-box attacks exhibit meaningful true positive rates (≥ 0.10) starting from 0.1% FPR, while on WikiMIA_2024 hard, these same attacks only reach this threshold at higher 1 or 5% FPR thresholds, if at all. This difference is most pronounced on smaller models like LLaMa-7B and 13B. Finally, for additional context, using AUROC, the best attack on WikiMIA_2024 Hard is lower than on WikiMIA on six out of seven comparable models, further indicating a reduction of easy temporal shortcuts.
> > > >
> > > > Overall, these results indicate that **WikiMIA-24's temporal distribution shift provides exploitable signals that WikiMIA_2024 Hard significantly mitigates.**  WikiMIA_2024 Hard offers a stricter and more realistic testbed for evaluating membership inference, reducing spurious cues and better isolating actual memorization behavior.
> > > >
> > > > ---
> > > >
> > > > **Concern 5** *What is the meaning of edit distance (Line 210)? Why should we ensure a sufficient distance between the member and non-member? The necessity of this operation is not justified.*
> > > >
> > > > Thank you for pointing this out – we agree that the justification for this design choice could be made clearer. **Edit distance** measures how much text has changed between two versions. In our case, we use **Levenshtein Edit Distance** [11], which counts the minimum number of single-character insertions, deletions, or substitutions needed to transform one text into another.
> > > >
> > > > In WikiMIA_2024 Hard, member/non-member pairs are constructed from different versions of the *same* Wikipedia article (pre-2017 and post-2024). However, many articles remain largely unchanged over time – some versions differ by only a few words. If we include such near-identical pairs, the task may become too difficult: even attacks targeting models that truly have memorized the older (member) version would struggle to distinguish the two, simply because the surface-level signal is too subtle to reliably detect. As a result, performance would become uniformly low, reducing the utility and interpretability of the benchmark.
> > > >
> > > > To avoid this, when creating WikiMIA_2024 Hard, we filter article pairs for a **minimum edit distance**, ensuring the newer (non-member) version differs meaningfully from the older (member) one. This makes the benchmark **challenging but not impossible**, so that observed model performance reflects actual memorization capability rather than the inability to detect imperceptible differences.
> > > >
> > > > **Empirically, even with the edit distance filtering, we already find WikiMIA_2024 Hard to be a challenging benchmark**: AUROC scores across baselines and target models are relatively low, with the significant majority of scores near the random baseline of 0.5. In addition, TPR starts extremely low across models and benchmarks at 0.1% FPR, and increases slowly as this FPR threshold is increased: even at 1% FPR, a large number of baselines have ≤ 0.05 TPR. Overall, these weak AUROC and TPR at low FPR results indicate that the benchmark is non-trivial as constructed. **Including even more similar pairs would likely push performance even closer to random**, eliminating our ability to meaningfully compare methods.
> > > >
> > > > **We will clarify edit distance and this motivation more explicitly in the next version of the paper.**

---

> > ### Author Response · Authors · 2025-06-03
> > **Response to Reviewer avj5 (Part 5)**
> >
> > # Updated Experiment Results
> > Below, we include updated versions of all main-text tables and the new WikiMIA-24, now expanded to include the additional metrics (TPR at multiple throw FPR thresholds). For clarity, tables are split up by dataset and metric. The first four columns (under “Ours →”) show our N-Gram Coverage Attack variants; the remaining columns (under “Baselines →”) correspond to the baselines. For each row, the highest-performing black-box method is **bolded**.
> >
> > ## BookMIA
> > ### AUROC
> >
> > | | Ours→ | | | | Baselines→ | | | | |
> > |---------------------|-------|----------|-------|----------|------------|------|--------|------|------|
> > | | Cov. | Cre. | LCS_C | LCS_W | DC | Loss | R-Loss | zlib | MinK |
> > | GPT-3.5-0125 | 0.84 | **0.85** | 0.84 | 0.83 | 0.84 | - | - | - | - |
> > | GPT-3.5 Inst. | 0.91 | 0.91 | 0.92 | **0.93** | 0.68 | - | - | - | - |
> > | GPT-3.5-1106 | 0.84 | **0.85** | 0.83 | 0.84 | **0.85** | - | - | - | - |
> >
> > ### TPR @ 0.1% FPR
> >
> > | | Ours→ | | | | Baselines→ | | | | |
> > |---------------------|-------|----------|-------|-------|------------|------|--------|------|------|
> > | | Cov. | Cre. | LCS_C | LCS_W | DC | Loss | R-Loss | zlib | MinK |
> > | GPT-3.5-0125 | 0.35 | **0.42** | 0.19 | 0.08 | 0.23 | - | - | - | - |
> > | GPT-3.5 Inst. | 0.51 | **0.52** | 0.38 | 0.42 | 0.04 | - | - | - | - |
> > | GPT-3.5-1106 | 0.35 | **0.39** | 0.16 | 0.05 | 0.18 | - | - | - | - |
> > ### TPR @ 0.5% FPR
> >
> > | | Ours→ | | | | Baselines→ | | | | |
> > |---------------------|-------|----------|-------|-------|------------|------|--------|------|------|
> > | | Cov. | Cre. | LCS_C | LCS_W | DC | Loss | R-Loss | zlib | MinK |
> > | GPT-3.5-0125 | 0.37 | **0.47** | 0.19 | 0.14 | 0.27 | - | - | - | - |
> > | GPT-3.5 Inst. | 0.56 | **0.59** | 0.50 | 0.44 | 0.08 | - | - | - | - |
> > | GPT-3.5-1106 | 0.51 | **0.58** | 0.30 | 0.08 | 0.37 | - | - | - | - |
> > ### TPR @ 1.0% FPR
> > | | Ours→ | | | | Baselines→ | | | | |
> > |---------------------|-------|----------|-------|-------|------------|------|--------|------|------|
> > | | Cov. | Cre. | LCS_C | LCS_W | DC | Loss | R-Loss | zlib | MinK |
> > | GPT-3.5-0125 | 0.38 | **0.49** | 0.23 | 0.17 | 0.33 | - | - | - | - |
> > | GPT-3.5 Inst. | 0.61 | **0.63** | 0.50 | 0.44 | 0.16 | - | - | - | - |
> > | GPT-3.5-1106 | 0.52 | **0.58** | 0.34 | 0.23 | 0.44 | - | - | - | - |
> >
> > ### TPR @ 5.0% FPR
> > | | Ours→ | | | | Baselines→ | | | | |
> > |---------------------|-------|----------|-------|-------|------------|------|--------|------|------|
> > | | Cov. | Cre. | LCS_C | LCS_W | DC | Loss | R-Loss | zlib | MinK |
> > | GPT-3.5-0125 | 0.52 | **0.59** | 0.52 | 0.48 | 0.54 | - | - | - | - |
> > | GPT-3.5 Inst. | 0.75 | **0.77** | 0.66 | 0.71 | 0.31 | - | - | - | - |
> > | GPT-3.5-1106 | 0.59 | **0.62** | 0.55 | 0.53 | 0.55 | - | - | - | - |
> >
> > ##  Tulu
> >
> > ### AUROC
> > | | Ours→ | | | | Baselines→ | | | | |
> > |--------------|----------|----------|-------|-------|------------|------|--------|------|------|
> > | | Cov. | Cre. | LCS_C | LCS_W | DC | Loss | R-Loss | zlib | MinK |
> > | Tulu-7B | **0.79** | **0.79** | 0.73 | 0.74 | 0.48 | 0.84 | 0.50 | 0.81 | 0.84 |
> > | Tulu-13B | **0.80** | **0.80** | 0.74 | 0.76 | 0.47 | 0.87 | 0.63 | 0.83 | 0.87 |
> > | Tulu-30B | **0.82** | **0.82** | 0.76 | 0.77 | 0.44 | 0.87 | 0.54 | 0.84 | 0.87 |
> > | Tulu-65B | 0.85 | **0.86** | 0.80 | 0.80 | 0.44 | 0.92 | 0.68 | 0.90 | 0.92 |
> > | Tulu-1.1-7B | 0.72 | **0.73** | 0.70 | 0.71 | 0.43 | 0.77 | 0.50 | 0.74 | 0.76 |
> > | Tulu-1.1-13B | **0.76** | 0.75 | 0.71 | 0.72 | 0.45 | 0.81 | 0.58 | 0.77 | 0.81 |
> > | Tulu-1.1-70B | **0.79** | 0.78 | 0.75 | 0.77 | 0.47 | 0.86 | 0.64 | 0.84 | 0.86 |
> >
> > ### TPR @ 0.1% FPR
> > |              | Ours→ |      |       |       | Baselines→ |      |        |      |      |
> > |--------------|-------|------|-------|-------|------------|------|--------|------|------|
> > |              | Cov.  | Cre. | LCS_C | LCS_W | DC         | Loss | R-Loss | zlib | MinK |
> > | Tulu-7B      | 0.00  | 0.00 | 0.00  | 0.00  | 0.00       | 0.00 | 0.00   | 0.00 | 0.00 |
> > | Tulu-13B     | 0.00  | 0.00 | 0.00  | 0.00  | 0.00       | 0.00 | 0.01   | 0.01 | 0.00 |
> > | Tulu-30B     | 0.00  | 0.00 | 0.00  | 0.00  | 0.00       | 0.01 | 0.00   | 0.01 | 0.01 |
> > | Tulu-65B     | 0.00  | 0.00 | 0.00  | 0.00  | 0.00       | 0.04 | 0.00   | 0.01 | 0.04 |
> > | Tulu-1.1-7B  | 0.00  | 0.00 | 0.00  | 0.00  | 0.00       | 0.00 | 0.00   | 0.00 | 0.00 |
> > | Tulu-1.1-13B | 0.00  | 0.00 | 0.00  | 0.00  | 0.00       | 0.01 | 0.00   | 0.00 | 0.01 |
> > | Tulu-1.1-70B | 0.00  | 0.00 | 0.00  | 0.00  | 0.00       | 0.01 | 0.00   | 0.03 | 0.01 |

---

> > > ### Author Response · Authors · 2025-06-03
> > > **Response to Reviewer avj5 (Part 6)**
> > >
> > > (continued Tulu tables)
> > > ### TPR @ 0.5% FPR
> > > | | Ours→ | | | | Baselines→ | | | | |
> > > |--------------|-------|------|----------|----------|------------|------|--------|------|------|
> > > | | Cov. | Cre. | LCS_C | LCS_W | DC | Loss | R-Loss | zlib | MinK |
> > > | Tulu-7B | 0.00 | 0.00 | 0.01 | **0.03** | 0.00 | 0.04 | 0.00 | 0.04 | 0.04 |
> > > | Tulu-13B | 0.00 | 0.00 | **0.12** | 0.05 | 0.00 | 0.04 | 0.02 | 0.06 | 0.04 |
> > > | Tulu-30B | 0.00 | 0.00 | **0.11** | 0.08 | 0.00 | 0.03 | 0.00 | 0.09 | 0.03 |
> > > | Tulu-65B | 0.00 | 0.00 | **0.12** | 0.10 | 0.00 | 0.09 | 0.00 | 0.22 | 0.09 |
> > > | Tulu-1.1-7B | 0.00 | 0.00 | **0.06** | 0.04 | 0 | 0.05 | 0.00 | 0.10 | 0.05 |
> > > | Tulu-1.1-13B | 0.00 | 0.00 | **0.05** | 0.05 | 0 | 0.03 | 0.00 | 0.18 | 0.03 |
> > > | Tulu-1.1-70B | 0.00 | 0.00 | **0.18** | 0.16 | 0 | 0.02 | 0.00 | 0.06 | 0.02 |
> > > ### TPR @ 1.0% FPR
> > > | | Ours→ | | | | Baselines→ | | | | |
> > > |--------------|-------|----------|----------|----------|------------|------|--------|------|------|
> > > | | Cov. | Cre. | LCS_C | LCS_W | DC | Loss | R-Loss | zlib | MinK |
> > > | Tulu-7B | 0.00 | 0.00 | **0.12** | 0.08 | 0.00 | 0.08 | 0.00 | 0.25 | 0.08 |
> > > | Tulu-13B | 0.00 | 0.16 | **0.17** | 0.14 | 0.00 | 0.13 | 0.02 | 0.28 | 0.13 |
> > > | Tulu-30B | 0.00 | 0.00 | **0.14** | 0.13 | 0.00 | 0.13 | 0.00 | 0.29 | 0.12 |
> > > | Tulu-65B | 0.00 | **0.23** | **0.23** | 0.22 | 0.00 | 0.24 | 0.01 | 0.44 | 0.24 |
> > > | Tulu-1.1-7B | 0.00 | 0.07 | **0.10** | 0.08 | 0.00 | 0.07 | 0.00 | 0.19 | 0.07 |
> > > | Tulu-1.1-13B | 0.00 | 0.08 | **0.11** | **0.11** | 0.00 | 0.12 | 0.01 | 0.24 | 0.12 |
> > > | Tulu-1.1-70B | 0.00 | 0.12 | 0.18 | **0.21** | 0.00 | 0.07 | 0.02 | 0.12 | 0.08 |
> > > ### TPR @ 5.0% FPR
> > > | | Ours→ | | | | Baselines→ | | | | |
> > > |--------------|-------|------|----------|----------|------------|------|--------|------|------|
> > > | | Cov. | Cre. | LCS_C | LCS_W | DC | Loss | R-Loss | zlib | MinK |
> > > | Tulu-7B | 0.00 | 0.27 | **0.35** | 0.32 | 0.03 | 0.40 | 0.00 | 0.43 | 0.41 |
> > > | Tulu-13B | 0.00 | 0.32 | 0.35 | **0.38** | 0.03 | 0.55 | 0.06 | 0.49 | 0.55 |
> > > | Tulu-30B | 0.00 | 0.35 | **0.39** | **0.39** | 0.00 | 0.45 | 0.03 | 0.50 | 0.45 |
> > > | Tulu-65B | 0.00 | 0.43 | 0.46 | **0.47** | 0.00 | 0.67 | 0.09 | 0.70 | 0.67 |
> > > | Tulu-1.1-7B | 0.00 | 0.22 | 0.29 | **0.31** | 0 | 0.36 | 0.00 | 0.36 | 0.36 |
> > > | Tulu-1.1-13B | 0.00 | 0.27 | 0.31 | **0.35** | 0 | 0.44 | 0.08 | 0.41 | 0.44 |
> > > | Tulu-1.1-70B | 0.00 | 0.33 | 0.37 | **0.40** | 0.04 | 0.43 | 0.12 | 0.54 | 0.44 |
> > >
> > > ## WikiMIA_2024 Hard
> > >
> > > ### AUROC
> > > | | Ours→ | | | | Baselines→ | | | | |
> > > |---------------|----------|----------|-------|-------|------------|------|--------|------|------|
> > > | | Cov. | Cre. | LCS_C | LCS_W | DC | Loss | R-Loss | zlib | MinK |
> > > | GPT-3.5-0125 | **0.59** | 0.57 | 0.55 | 0.55 | 0.47 | - | - | - | - |
> > > | GPT-3.5 Inst. | **0.64** | 0.63 | 0.61 | 0.61 | 0.45 | - | - | - | - |
> > > | GPT-3.5-1106 | **0.58** | **0.58** | 0.56 | 0.57 | 0.49 | - | - | - | - |
> > > | GPT-4 | 0.57 | **0.58** | 0.55 | 0.57 | 0.44 | - | - | - | - |
> > > | GPT-4o-1120 | **0.55** | **0.55** | 0.54 | 0.52 | 0.51 | - | - | - | - |
> > > | GPT-4o Mini | **0.55** | 0.53 | 0.52 | 0.52 | 0.43 | - | - | - | - |
> > > | LLaMA-7B | **0.55** | 0.54 | 0.57 | 0.54 | 0.47 | - | - | 0.50 | 0.52 |
> > > | LLaMA-13B | **0.59** | 0.58 | 0.57 | 0.54 | 0.51 | 0.53 | 0.57 | 0.51 | 0.54 |
> > > | LLaMA-30B | **0.61** | **0.61** | 0.60 | 0.57 | 0.50 | 0.56 | 0.61 | 0.53 | 0.60 |
> > > | LLaMA-65B | **0.64** | 0.63 | 0.60 | 0.60 | 0.51 | 0.57 | 0.61 | 0.54 | 0.58 |
> > > ### TPR @ 0.1% FPR
> > > | | Ours→ | | | | Baselines→ | | | | |
> > > |---------------|----------|----------|----------|----------|------------|------|--------|------|------|
> > > | | Cov. | Cre. | LCS_C | LCS_W | DC | Loss | R-Loss | zlib | MinK |
> > > | GPT-3.5-0125 | 0.02 | 0.01 | **0.03** | 0.00 | 0.00 | - | - | - | - |
> > > | GPT-3.5 Inst. | 0.03 | **0.05** | 0.02 | 0.01 | 0.01 | - | - | - | - |
> > > | GPT-3.5-1106 | 0.02 | 0.02 | 0.05 | **0.06** | 0.03 | - | - | - | - |
> > > | GPT-4 | 0.01 | 0.02 | **0.04** | **0.04** | 0.00 | - | - | - | - |
> > > | GPT-4o-1120 | **0.02** | **0.02** | 0.01 | 0.01 | 0.00 | - | - | - | - |
> > > | GPT-4o Mini | **0.04** | 0.02 | 0.01 | 0.01 | 0.00 | - | - | - | - |
> > > | LLaMA-7B | 0.00 | 0.02 | **0.05** | 0.01 | 0.00 | 0.03 | - | 0.00 | 0.01 |
> > > | LLaMA-13B | 0.00 | 0.01 | **0.02** | **0.02** | 0.01 | 0.03 | 0.04 | 0.03 | 0.03 |
> > > | LLaMA-30B | 0.00 | 0.05 | **0.05** | 0.04 | 0.00 | 0.14 | 0.03 | 0.10 | 0.04 |
> > > | LLaMA-65B | 0.00 | 0.08 | 0.07 | **0.10** | 0.00 | 0.16 | 0.04 | 0.12 | 0.09 |

---

> > > > ### Author Response · Authors · 2025-06-03
> > > > **Response to Reviewer avj5 (Part 7)**
> > > >
> > > > (continued WikiMIA_2024 Hard Tables)
> > > >
> > > > ### TPR @ 0.5% FPR
> > > > | | Ours→ | | | | Baselines→ | | | | |
> > > > |---------------|----------|----------|----------|----------|------------|------|--------|------|------|
> > > > | | Cov. | Cre. | LCS_C | LCS_W | DC | Loss | R-Loss | zlib | MinK |
> > > > | GPT-3.5-0125 | 0.02 | 0.01 | **0.03** | 0.00 | 0.00 | - | - | - | - |
> > > > | GPT-3.5 Inst. | 0.03 | **0.05** | 0.02 | 0.01 | 0.01 | - | - | - | - |
> > > > | GPT-3.5-1106 | 0.02 | 0.02 | 0.05 | **0.06** | 0.03 | - | - | - | - |
> > > > | GPT-4 | 0.01 | 0.02 | **0.04** | **0.04** | 0.00 | - | - | - | - |
> > > > | GPT-4o-1120 | **0.02** | **0.02** | 0.01 | 0.01 | 0.00 | - | - | - | - |
> > > > | GPT-4o Mini | **0.04** | 0.02 | 0.01 | 0.01 | 0.00 | - | - | - | - |
> > > > | LLaMA-7B | 0.00 | 0.02 | **0.05** | 0.01 | 0.00 | 0.03 | - | 0.00 | 0.01 |
> > > > | LLaMA-13B | 0.00 | 0.01 | **0.02** | **0.02** | 0.01 | 0.03 | 0.04 | 0.03 | 0.03 |
> > > > | LLaMA-30B | 0.00 | 0.05 | **0.05** | 0.04 | 0.00 | 0.14 | 0.03 | 0.10 | 0.04 |
> > > > | LLaMA-65B | 0.00 | 0.08 | 0.07 | **0.10** | 0.00 | 0.16 | 0.04 | 0.12 | 0.09 |
> > > > ### TPR @ 1.0% FPR
> > > > | | Ours→ | | | | Baselines→ | | | | |
> > > > |---------------|----------|----------|----------|----------|------------|------|--------|------|------|
> > > > | | Cov. | Cre. | LCS_C | LCS_W | DC | Loss | R-Loss | zlib | MinK |
> > > > | GPT-3.5-0125 | 0.02 | 0.01 | 0.03 | **0.04** | 0.02 | - | - | - | - |
> > > > | GPT-3.5 Inst. | 0.05 | **0.06** | 0.04 | **0.06** | 0.01 | - | - | - | - |
> > > > | GPT-3.5-1106 | 0.04 | **0.08** | 0.07 | 0.06 | 0.03 | - | - | - | - |
> > > > | GPT-4 | 0.01 | 0.04 | 0.05 | **0.08** | 0.00 | - | - | - | - |
> > > > | GPT-4o-1120 | **0.03** | 0.02 | 0.02 | **0.03** | 0.00 | - | - | - | - |
> > > > | GPT-4o Mini | **0.04** | 0.03 | 0.02 | 0.01 | 0.00 | - | - | - | - |
> > > > | LLaMA-7B | 0.04 | 0.04 | **0.05** | 0.01 | 0.03 | 0.03 | - | 0.02 | 0.04 |
> > > > | LLaMA-13B | 0.00 | 0.02 | 0.04 | 0.02 | **0.05** | 0.04 | 0.05 | 0.04 | 0.03 |
> > > > | LLaMA-30B | 0.00 | 0.05 | **0.09** | 0.04 | 0.00 | 0.16 | 0.07 | 0.11 | 0.09 |
> > > > | LLaMA-65B | 0.00 | 0.09 | 0.08 | **0.10** | 0.00 | 0.18 | 0.09 | 0.13 | 0.10 |
> > > > ### TPR @ 5.0% FPR
> > > > | | Ours→ | | | | Baselines→ | | | | |
> > > > |---------------|----------|----------|----------|----------|------------|------|--------|------|------|
> > > > | | Cov. | Cre. | LCS_C | LCS_W | DC | Loss | R-Loss | zlib | MinK |
> > > > | GPT-3.5-0125 | **0.10** | 0.09 | 0.06 | 0.09 | 0.06 | - | - | - | - |
> > > > | GPT-3.5 Inst. | 0.12 | 0.14 | 0.13 | **0.15** | 0.01 | - | - | - | - |
> > > > | GPT-3.5-1106 | 0.09 | 0.11 | 0.10 | **0.15** | 0.04 | - | - | - | - |
> > > > | GPT-4 | 0.10 | 0.05 | **0.13** | 0.11 | 0.00 | - | - | - | - |
> > > > | GPT-4o-1120 | 0.09 | **0.13** | 0.10 | 0.06 | 0.00 | - | - | - | - |
> > > > | GPT-4o Mini | **0.08** | 0.07 | 0.02 | 0.06 | 0.04 | - | - | - | - |
> > > > | LLaMA-7B | 0.10 | **0.17** | 0.06 | 0.03 | 0.03 | 0.08 | - | 0.07 | 0.09 |
> > > > | LLaMA-13B | 0.00 | **0.15** | 0.08 | 0.05 | 0.06 | 0.10 | 0.11 | 0.09 | 0.06 |
> > > > | LLaMA-30B | 0.00 | 0.09 | **0.13** | **0.13** | 0.05 | 0.20 | 0.17 | 0.13 | 0.16 |
> > > > | LLaMA-65B | **0.24** | 0.14 | 0.13 | 0.13 | 0.02 | 0.20 | 0.21 | 0.14 | 0.18 |
> > > > ## WikiMIA
> > > > ### AUROC
> > > > | | Ours→ | | | | Baselines→ | | | | |
> > > > |---------------|----------|------|-------|-------|------------|------|--------|------|------|
> > > > | | Cov. | Cre. | LCS_C | LCS_W | DC | Loss | R-Loss | zlib | MinK |
> > > > | GPT-3.5-0125 | **0.64** | 0.63 | 0.61 | 0.60 | 0.54 | - | - | - | - |
> > > > | GPT-3.5 Inst. | **0.62** | 0.61 | 0.58 | 0.58 | 0.54 | - | - | - | - |
> > > > | GPT-3.5-1106 | **0.64** | 0.62 | 0.61 | 0.60 | 0.52 | - | - | - | - |
> > > > | LLaMA-7B | **0.60** | 0.59 | 0.56 | 0.55 | 0.48 | 0.62 | - | 0.63 | 0.64 |
> > > > | LLaMA-13B | **0.62** | 0.59 | 0.57 | 0.54 | 0.52 | 0.64 | 0.63 | 0.65 | 0.66 |
> > > > | LLaMA-30B | **0.63** | 0.62 | 0.57 | 0.58 | 0.49 | 0.66 | 0.69 | 0.67 | 0.69 |
> > > > | LLaMA-65B | **0.65** | 0.64 | 0.61 | 0.58 | 0.50 | 0.68 | 0.74 | 0.69 | 0.70 |
> > > > ### TPR @ 0.1% FPR
> > > > | | Ours→ | | | | Baselines→ | | | | |
> > > > |---------------|----------|----------|----------|----------|------------|------|--------|------|------|
> > > > | | Cov. | Cre. | LCS_C | LCS_W | DC | Loss | R-Loss | zlib | MinK |
> > > > | GPT-3.5-0125 | 0.00 | 0.00 | 0.00 | 0.00 | **0.03** | - | - | - | - |
> > > > | GPT-3.5 Inst. | 0.00 | **0.02** | 0.00 | 0.01 | 0.00 | - | - | - | - |
> > > > | GPT-3.5-1106 | 0.00 | 0.00 | 0.00 | **0.02** | 0.02 | - | - | - | - |
> > > > | LLaMA-7B | **0.02** | 0.00 | 0.00 | 0.00 | 0.00 | 0.04 | - | 0.04 | 0.04 |
> > > > | LLaMA-13B | 0.00 | 0.00 | 0.00 | 0.00 | **0.01** | 0.04 | 0.00 | 0.07 | 0.05 |
> > > > | LLaMA-30B | 0.00 | 0.00 | **0.01** | 0.00 | 0.00 | 0.03 | 0.00 | 0.04 | 0.04 |
> > > > | LLaMA-65B | 0.00 | **0.02** | 0.01 | **0.02** | 0.01 | 0.03 | 0.00 | 0.03 | 0.03 |

---

> > ### Author Response · Authors · 2025-06-03
> > **Response to Reviewer avj5 (Part 8)**
> >
> > (continued WikiMIA tables)
> > ### TPR @ 0.5% FPR
> > | | Ours→ | | | | Baselines→ | | | | |
> > |---------------|----------|----------|-------|----------|------------|------|--------|------|------|
> > | | Cov. | Cre. | LCS_C | LCS_W | DC | Loss | R-Loss | zlib | MinK |
> > | GPT-3.5-0125 | **0.04** | 0.00 | 0.00 | 0.01 | 0.03 | - | - | - | - |
> > | GPT-3.5 Inst. | **0.02** | **0.02** | 0.01 | 0.01 | 0.00 | - | - | - | - |
> > | GPT-3.5-1106 | 0.01 | 0.02 | 0.02 | 0.02 | **0.03** | - | - | - | - |
> > | LLaMA-7B | **0.02** | 0.00 | 0.00 | 0.00 | 0.00 | 0.04 | - | 0.06 | 0.05 |
> > | LLaMA-13B | 0.01 | 0.01 | 0.00 | 0.01 | **0.04** | 0.05 | 0.01 | 0.09 | 0.06 |
> > | LLaMA-30B | 0.00 | 0.01 | 0.03 | 0.01 | **0.05** | 0.06 | 0.00 | 0.10 | 0.08 |
> > | LLaMA-65B | 0.00 | **0.03** | 0.01 | **0.03** | 0.01 | 0.10 | 0.00 | 0.13 | 0.12 |
> > ### TPR @ 1.0% FPR
> > | | Ours→ | | | | Baselines→ | | | | |
> > |---------------|----------|----------|----------|----------|------------|------|--------|------|------|
> > | | Cov. | Cre. | LCS_C | LCS_W | DC | Loss | R-Loss | zlib | MinK |
> > | GPT-3.5-0125 | **0.05** | 0.02 | 0.02 | 0.01 | **0.05** | - | - | - | - |
> > | GPT-3.5 Inst. | 0.02 | **0.03** | 0.01 | 0.01 | 0.02 | - | - | - | - |
> > | GPT-3.5-1106 | 0.04 | 0.03 | 0.02 | 0.02 | **0.05** | - | - | - | - |
> > | LLaMA-7B | **0.02** | **0.02** | **0.02** | **0.02** | 0.00 | 0.06 | - | 0.08 | 0.08 |
> > | LLaMA-13B | 0.01 | 0.01 | 0.00 | 0.02 | **0.04** | 0.08 | 0.03 | 0.09 | 0.08 |
> > | LLaMA-30B | 0.02 | 0.01 | 0.04 | 0.01 | **0.05** | 0.07 | 0.02 | 0.10 | 0.08 |
> > | LLaMA-65B | 0.01 | **0.03** | 0.02 | **0.03** | 0.01 | 0.10 | 0.00 | 0.14 | 0.13 |
> > ### TPR @ 5.0% FPR
> > | | Ours→ | | | | Baselines→ | | | | |
> > |---------------|----------|----------|-------|-------|------------|------|--------|------|------|
> > | | Cov. | Cre. | LCS_C | LCS_W | DC | Loss | R-Loss | zlib | MinK |
> > | GPT-3.5-0125 | 0.11 | 0.10 | 0.06 | 0.07 | **0.15** | - | - | - | - |
> > | GPT-3.5 Inst. | **0.11** | 0.09 | 0.07 | 0.08 | 0.07 | - | - | - | - |
> > | GPT-3.5-1106 | **0.13** | **0.13** | 0.08 | 0.10 | 0.11 | - | - | - | - |
> > | LLaMA-7B | **0.10** | 0.08 | 0.06 | 0.05 | 0.02 | 0.08 | - | 0.12 | 0.17 |
> > | LLaMA-13B | **0.08** | 0.05 | 0.05 | 0.04 | 0.06 | 0.13 | 0.04 | 0.14 | 0.18 |
> > | LLaMA-30B | 0.08 | **0.11** | 0.05 | 0.10 | 0.10 | 0.15 | 0.11 | 0.17 | 0.18 |
> > | LLaMA-65B | 0.10 | **0.11** | 0.09 | 0.07 | 0.01 | 0.17 | 0.14 | 0.18 | 0.20 |
> >
> > ## WikiMIA-24 [1]
> > ### AUROC
> > | | Ours→ | | | | Baselines→ | | | | |
> > |---------------|----------|----------|----------|----------|------------|------|--------|------|------|
> > | | Cov. | Cre. | LCS_C | LCS_W | DC | Loss | R-Loss | zlib | MinK |
> > | GPT-3.5-0125 | **0.67** | **0.67** | 0.64 | 0.66 | 0.48 | - | - | - | - |
> > | GPT-3.5 Inst. | **0.65** | 0.64 | 0.62 | 0.64 | 0.50 | - | - | - | - |
> > | GPT-3.5-1106 | **0.68** | 0.67 | 0.66 | 0.68 | 0.49 | - | - | - | - |
> > | GPT-4 | **0.84** | 0.82 | 0.76 | 0.79 | 0.56 | - | - | - | - |
> > | GPT-4o-1120 | **0.83** | 0.82 | 0.77 | 0.79 | 0.50 | - | - | - | - |
> > | GPT-4o Mini | 0.73 | **0.74** | 0.66 | 0.69 | 0.44 | - | - | - | - |
> > | LLaMA-7B | 0.59 | 0.59 | **0.60** | 0.59 | 0.53 | 0.67 | - | 0.67 | 0.69 |
> > | LLaMA-13B | **0.63** | **0.63** | 0.61 | 0.61 | 0.50 | 0.68 | 0.60 | 0.69 | 0.71 |
> > | LLaMA-30B | **0.67** | 0.66 | 0.64 | 0.64 | 0.48 | 0.72 | 0.69 | 0.72 | 0.74 |
> > | LLaMA-65B | 0.64 | **0.65** | **0.65** | **0.65** | 0.50 | 0.74 | 0.74 | 0.75 | 0.76 |
> > ### TPR @ 0.1% FPR
> > | | Ours→ | | | | Baselines→ | | | | |
> > |---------------|----------|----------|----------|----------|------------|------|--------|------|------|
> > | | Cov. | Cre. | LCS_C | LCS_W | DC | Loss | R-Loss | zlib | MinK |
> > | GPT-3.5-0125 | 0.00 | 0.00 | 0.01 | **0.02** | 0.00 | - | - | - | - |
> > | GPT-3.5 Inst. | **0.01** | 0.00 | 0.00 | **0.01** | **0.01** | - | - | - | - |
> > | GPT-3.5-1106 | 0.02 | 0.02 | **0.03** | 0.02 | 0.00 | - | - | - | - |
> > | GPT-4 | **0.09** | 0.07 | 0.03 | 0.03 | 0.00 | - | - | - | - |
> > | GPT-4o-1120 | **0.05** | 0.03 | 0.01 | **0.05** | 0.00 | - | - | - | - |
> > | GPT-4o Mini | 0.04 | **0.07** | 0.06 | 0.06 | 0.00 | - | - | - | - |
> > | LLaMA-7B | 0.03 | 0.04 | 0.00 | **0.06** | 0.00 | 0.14 | - | 0.13 | 0.17 |
> > | LLaMA-13B | 0.00 | 0.00 | 0.01 | **0.03** | 0.00 | 0.07 | 0.02 | 0.12 | 0.07 |
> > | LLaMA-30B | 0.00 | 0.02 | 0.02 | 0.02 | **0.03** | 0.13 | 0.05 | 0.15 | 0.16 |
> > | LLaMA-65B | 0.01 | 0.00 | 0.01 | **0.03** | 0.00 | 0.09 | 0.08 | 0.09 | 0.15 |

---

> > ### Author Response · Authors · 2025-06-03
> > **Response to Reviewer avj5 (Part 9)**
> >
> > (continued WikiMIA-24 tables)
> > ### TPR @ 0.5% FPR
> > | | Ours→ | | | | Baselines→ | | | | |
> > |---------------|----------|----------|-------|----------|------------|------|--------|------|------|
> > | | Cov. | Cre. | LCS_C | LCS_W | DC | Loss | R-Loss | zlib | MinK |
> > | GPT-3.5-0125 | 0.01 | 0.00 | 0.02 | **0.03** | 0.00 | - | - | - | - |
> > | GPT-3.5 Inst. | **0.02** | **0.02** | 0.00 | 0.01 | 0.01 | - | - | - | - |
> > | GPT-3.5-1106 | 0.03 | **0.06** | 0.03 | 0.02 | 0.00 | - | - | - | - |
> > | GPT-4 | 0.16 | **0.21** | 0.03 | 0.04 | 0.00 | - | - | - | - |
> > | GPT-4o-1120 | **0.08** | 0.07 | 0.05 | 0.06 | 0.00 | - | - | - | - |
> > | GPT-4o Mini | **0.08** | **0.08** | 0.06 | 0.06 | 0.00 | - | - | - | - |
> > | LLaMA-7B | 0.05 | 0.05 | 0.01 | **0.07** | 0.00 | 0.15 | - | 0.16 | 0.17 |
> > | LLaMA-13B | 0.00 | 0.00 | 0.03 | **0.04** | 0.00 | 0.14 | 0.02 | 0.14 | 0.20 |
> > | LLaMA-30B | 0.01 | **0.08** | 0.02 | 0.02 | 0.03 | 0.13 | 0.08 | 0.15 | 0.16 |
> > | LLaMA-65B | 0.01 | **0.06** | 0.05 | 0.03 | 0.02 | 0.11 | 0.08 | 0.10 | 0.16 |
> > ### TPR @ 1.0% FPR
> > | | Ours→ | | | | Baselines→ | | | | |
> > |---------------|----------|----------|-------|----------|------------|------|--------|------|------|
> > | | Cov. | Cre. | LCS_C | LCS_W | DC | Loss | R-Loss | zlib | MinK |
> > | GPT-3.5-0125 | 0.02 | 0.02 | 0.02 | **0.03** | 0.00 | - | - | - | - |
> > | GPT-3.5 Inst. | 0.02 | **0.03** | 0.02 | 0.01 | 0.02 | - | - | - | - |
> > | GPT-3.5-1106 | **0.06** | **0.06** | 0.03 | **0.06** | 0.00 | - | - | - | - |
> > | GPT-4 | 0.19 | **0.23** | 0.11 | 0.09 | 0.00 | - | - | - | - |
> > | GPT-4o-1120 | **0.10** | 0.07 | 0.06 | 0.09 | 0.00 | - | - | - | - |
> > | GPT-4o Mini | **0.08** | **0.08** | 0.06 | **0.08** | 0.00 | - | - | - | - |
> > | LLaMA-7B | 0.06 | **0.07** | 0.01 | **0.07** | 0.00 | 0.16 | - | 0.17 | 0.17 |
> > | LLaMA-13B | 0.00 | **0.04** | 0.03 | **0.04** | 0.01 | 0.14 | 0.03 | 0.14 | 0.21 |
> > | LLaMA-30B | 0.05 | **0.09** | 0.03 | 0.06 | 0.06 | 0.13 | 0.08 | 0.16 | 0.17 |
> > | LLaMA-65B | 0.06 | **0.12** | 0.09 | 0.06 | 0.02 | 0.13 | 0.11 | 0.10 | 0.16 |
> > ### TPR @ 5.0% FPR
> > | | Ours→ | | | | Baselines→ | | | | |
> > |---------------|----------|----------|-------|----------|------------|------|--------|------|------|
> > | | Cov. | Cre. | LCS_C | LCS_W | DC | Loss | R-Loss | zlib | MinK |
> > | GPT-3.5-0125 | **0.18** | 0.15 | 0.14 | 0.12 | 0.05 | - | - | - | - |
> > | GPT-3.5 Inst. | 0.10 | **0.16** | 0.12 | 0.08 | 0.03 | - | - | - | - |
> > | GPT-3.5-1106 | **0.20** | 0.19 | 0.16 | 0.19 | 0.03 | - | - | - | - |
> > | GPT-4 | 0.48 | **0.51** | 0.30 | 0.31 | 0.00 | - | - | - | - |
> > | GPT-4o-1120 | **0.29** | 0.25 | 0.18 | 0.22 | 0.00 | - | - | - | - |
> > | GPT-4o Mini | **0.22** | 0.19 | 0.14 | 0.15 | 0.02 | - | - | - | - |
> > | LLaMA-7B | **0.11** | 0.09 | 0.08 | 0.10 | 0.03 | 0.23 | - | 0.25 | 0.25 |
> > | LLaMA-13B | **0.16** | 0.12 | 0.12 | 0.14 | 0.07 | 0.27 | 0.18 | 0.33 | 0.31 |
> > | LLaMA-30B | 0.14 | 0.17 | 0.22 | **0.25** | 0.03 | 0.34 | 0.23 | 0.36 | 0.36 |
> > | LLaMA-65B | 0.16 | **0.22** | 0.21 | 0.20 | 0.04 | 0.23 | 0.27 | 0.24 | 0.23 |

---

> > > ### Author Response · Authors · 2025-06-03
> > > **Response to Reviewer avj5 (Part 10)**
> > >
> > > # References
> > > [1]	Fu, Wenjie, et al. MIA-Tuner: Adapting Large Language Models as Pre-training Text Detector. Proceedings of the AAAI Conference on Artificial Intelligence, vol. 39, 2025, pp. 1234–1242. AAAI Press.
> > >
> > > [2]	Garg, Isha, et al. “Memorization through the Lens of Curvature of Loss Function around Samples.” Proceedings of the 41st International Conference on Machine Learning, PMLR, 2024.
> > >
> > > [3]	Ravikumar, Deepak, et al. “Unveiling Privacy, Memorization, and Input Curvature Links.” Proceedings of the 41st International Conference on Machine Learning, PMLR, 2024.
> > >
> > >  [4]	Zhang, Jingyang, et al. Min-K%++: Improved Baseline for Detecting Pre-Training Data from Large Language Models. arXiv:2404.02936, arXiv, 23 May 2024.
> > >
> > > [5]	Fu, Wenjie, et al. "Membership Inference Attacks against Fine-tuned Large Language Models via Self-prompt Calibration." The Thirty-eighth Annual Conference on Neural Information Processing Systems. 2024.
> > >
> > > [6]	Carlini, Nicholas, et al. "Membership inference attacks from first principles." 2022 IEEE Symposium on Security and Privacy (2022).
> > >
> > > [7]	Bertran, Martin, et al. "Scalable membership inference attacks via quantile regression." Advances in Neural Information Processing Systems (2023).
> > >
> > > [8] Wen, Yuxin, et al. Canary in a Coalmine: Better Membership Inference with Ensembled Adversarial Queries. International Conference on Learning Representations (ICLR), 2024.
> > >
> > > [9] Das, Dipanjan, et al. Blind Baselines Beat Membership Inference Attacks for Foundation Models. arXiv, 2024, arXiv:2406.16201.
> > >
> > > [10] Shi, Weijia, et al. Detecting Pretraining Data from Large Language Models. arXiv, 2023, arXiv:2310.16789.
> > >
> > > [11] Levenshtein, Vladimir I.. “Binary codes capable of correcting deletions, insertions, and reversals.” Soviet physics. Doklady 10 (1965): 707-710.
> > >
> > > [12] Duan, Michael et al. “Do Membership Inference Attacks Work on Large Language Models?” ArXiv abs/2402.07841 (2024)
> > >
> > > [13] Xie, Roy et al. “ReCaLL: Membership Inference via Relative Conditional Log-Likelihoods.” Conference on Empirical Methods in Natural Language Processing (2024).
> > >
> > > [14] Tran, Toan et al. “Tokens for Learning, Tokens for Unlearning: Mitigating Membership Inference Attacks in Large Language Models via Dual-Purpose Training.” ArXiv abs/2502.19726 (2025)
> > >
> > >
> > > [15] Liu, Mingrui et al. “Mask-based Membership Inference Attacks for Retrieval-Augmented Generation.” Proceedings of the ACM on Web Conference 2025 (2024)
> > >
> > > [16] Kim, Gyuwan et al. “Detecting Training Data of Large Language Models via Expectation Maximization.” ArXiv abs/2410.07582 (2024)
> > >
> > > [17] Chang, Hongyan et al. “Context-Aware Membership Inference Attacks against Pre-trained Large Language Models.” ArXiv abs/2409.13745 (2024)
> > >
> > > [18] Huang, Zhiheng et al. “DF-MIA: A Distribution-Free Membership Inference Attack on Fine-Tuned Large Language Models.” AAAI Conference on Artificial Intelligence (2025).
> > >
> > > [19] Naseh, Ali et al. “Riddle Me This! Stealthy Membership Inference for Retrieval-Augmented Generation.” ArXiv abs/2502.00306 (2025)
> > >
> > > [20] Antebi, Sagiv et al. “Tag&Tab: Pretraining Data Detection in Large Language Models Using Keyword-Based Membership Inference Attack.” ArXiv abs/2501.08454 (2025)
> > >
> > > [21] Mattern, Justus et al. “Membership Inference Attacks against Language Models via Neighbourhood Comparison.” ArXiv abs/2305.18462 (2023)
> > >
> > > [22] Panda, Ashwinee et al. “Privacy Auditing of Large Language Models.” ArXiv abs/2503.06808 (2025)
> > >
> > > [23] Zhang, Anqi and Chaofeng Wu. “Adaptive Pre-training Data Detection for Large Language Models via Surprising Tokens.” ArXiv abs/2407.21248 (2024): n. pag.
> > >
> > > [24] Wang, Cheng et al. “Con-ReCall: Detecting Pre-training Data in LLMs via Contrastive Decoding.” ArXiv abs/2409.03363 (2024)
> > >
> > > [25] Eichler, Cédric et al. “Nob-MIAs: Non-biased Membership Inference Attacks Assessment on Large Language Models with Ex-Post Dataset Construction.” WISE (2024).

---

> > > > ### Author Response · Authors · 2025-06-05
> > > > **Rebuttal Follow-Up**
> > > >
> > > > Thank you again for your time and valuable feedback on our paper.
> > > >
> > > > We wanted to gently follow up on our rebuttal. We are eager to know if our responses have addressed your concerns and would be happy to provide any further clarifications.
> > > >
> > > > Please let us know if you have any follow-up questions.

---

> > > > > ### Author Response · Authors · 2025-06-09
> > > > > **Request for Rebuttal Follow-Up**
> > > > >
> > > > > Thank you again for your time and valuable feedback on our paper.
> > > > >
> > > > > We wanted to gently follow up again on our rebuttal, given that the discussion period deadline is approaching. We are eager to know if our responses have addressed your concerns and would be happy to provide any further clarifications.
> > > > >
> > > > > Please let us know if you have any follow-up questions.

---

### Official Review · Reviewer_Yujw · 2025-05-13

**Rating:** 6
**Confidence:** 3
**Ethics Flag:** 1

**Summary:**

The paper presents a membership inference attack for black-box models. The main idea is to sample from the model using a prefix and compute similarity metrics between the model's completion and the ground truth. A threshold on these similarity scores is then used to detect whether a sample was part of the training data.

**Reasons To Accept:**

The paper addresses a timely and important problem: detecting potential copyright infringements in a restrictive setting where only output tokens (without probabilities) are available from the LLM's API.

The proposed method appears nearly as effective as white-box attacks. However, the overall evaluation could be improved, as discussed below.

**Reasons To Reject:**

There are several papers on black-box membership inference attacks on LLMs (such as [1], [2], and [3]) that deserve a (more) detailed discussion. These works share some similarities with the current approach, so a comparative analysis of results would be valuable.

While the authors introduced a new version of WikiMIA to mitigate distribution shift, other benchmarks used in the paper, such as BookMIA, are known to suffer from such shifts [4]. Incorporating benchmarks like MIMIR [5] could further validate the robustness of the method.

It would also be beneficial for the authors to evaluate their method using True Positive Rate (TPR) at a low False Positive Rate (FPR), which is a commonly used metric in membership inference.

Although the authors compare their method with [6], it's worth noting that [6] is designed for document-level membership inference, where results from multiple paragraphs are aggregated. In contrast, the black-box methods referenced earlier ([1]–[3]) might serve as more appropriate baselines for comparison.

[1] https://arxiv.org/pdf/2310.13771

[2] https://arxiv.org/pdf/2409.13831

[3] https://arxiv.org/pdf/2404.11262

[4] https://arxiv.org/pdf/2406.16201

[5] https://arxiv.org/pdf/2402.07841

[6] https://arxiv.org/pdf/2402.09910

---

> ### Author Response · Authors · 2025-06-03
> **Response to Reviewer Yujw (Part 1)**
>
> Thank you for your review! We appreciate that you felt our paper was relevant and “addresses a timely and important problem”. We are also glad you recognized the effectiveness of our proposed method, which is competitive even to white-box baselines.
>
> We address all your questions and concerns below, and would be happy to discuss any remaining points during the discussion period.
>
> ---
>
> **Concern 1**: *While the authors introduced a new version of WikiMIA to mitigate distribution shift, other benchmarks used in the paper, such as BookMIA, are known to suffer from such shifts [4]. Incorporating benchmarks like MIMIR [5] could further validate the robustness of the method.*
>
> Thank you for the suggestion! We appreciate your recognition that our new version of WikiMIA “mitigates distribution shift”. We address both these points below.
>
> **BookMIA and WikiMIA distribution shift**
>
> Regarding BookMIA and the original WikiMIA, we agree they have known distribution shift issues as mentioned by [4] - **we will more clearly mention this in the next version of the paper**, and will explicitly state “*Both BookMIA and the original WikiMIA have known distribution shift issues, as previously discussed in [4].*” However, given that these are among the only available benchmarks that can be used for closed-access OpenAI models (due to a lack of publicly-known training data) [9], we believe it is important to include them, even with their known limitations, to enable broader evaluation and comparison.
>
> In addition, to more thoroughly evaluate attacks, **we test on two other datasets with reduced distribution shift:** **WikiMIA_2024 Hard**, which provides a cleaner benchmark than WikiMIA, by sourcing members and non-members from different versions of the *same* article rather than entirely different articles, and **TULU**, whose members and non-members are drawn both from instruction-tuning datasets, ensuring they are topically aligned despite being sourced from different corpora; we also curate these datasets so that member and non-member data lengths are similar to reduce the possibility of length-based shortcuts. **Our results show strong performance across all datasets – both with and without distribution shift** – demonstrating that our method's effectiveness is not contingent on temporal artifacts. We believe this comprehensive evaluation across four datasets with varying levels of distribution shift provides useful evidence for the method’s robustness.
>
> **Suggestion on testing on MIMIR**
>
> Regarding MIMIR, we appreciate this suggestion and actually **already include this benchmark in our original paper!** In the Appendix, section "Additional Experiments" (page 16, line 605), we test on two datasets: the Pile dataset from MIMIR [7] and Dolma [8]. Lines 634-636 and the corresponding footnote explicitly confirm that we use the MIMIR-curated version of the Pile dataset. The results of these datasets are in Table 4 and 5 (Page 17). The same conclusions from the main text hold: our method N-Gram Coverage Attack performs better than DE-COP and comparatively to the white-box baselines, confirming that our main findings generalize beyond the four primary benchmarks in the main text. We hope these strong results sufficiently validate the robustness of our method.
>
> ---
>
> **Concern 2**: *It would also be beneficial for the authors to evaluate their method using True Positive Rate (TPR) at a low False Positive Rate (FPR), which is a commonly used metric in membership inference.*
>
> Thank you for this suggestion! We agree that TPR at low FPR is another important metric for membership inference. **In our updated experiments, we now report TPR@ 0.1, 0.5, 1.0, and 5.0% FPR alongside AUROC for all methods and datasets from the main text.** Overall, our method excels on both AUROC and TPR at low FPR, validating its effectiveness across different evaluation paradigms. Please see **Updated Experiments** and **Summary of Results** sections below.

---

> > ### Author Response · Authors · 2025-06-03
> > **Response to Reviewer Yujw (Part 2)**
> >
> > ---
> >
> > **Concern 3**: *There are several papers on black-box membership inference attacks on LLMs (such as [1], [2], and [3]) that deserve a (more) detailed discussion. These works share some similarities with the current approach, so a comparative analysis of results would be valuable.*
> >
> > …
> >
> > *Although the authors compare their method with [6], it's worth noting that [6] is designed for document-level membership inference, where results from multiple paragraphs are aggregated. In contrast, the black-box methods referenced earlier ([1]–[3]) might serve as more appropriate baselines for comparison.*
> >
> > Thank you for this suggestion! We agree that a more thorough comparison with recent black-box MIA approaches would strengthen our paper. We also agree that [6] is designed more for document-level inference, but included it as a baseline due to its strong performance and relevance as a recent black-box method. That said, we now additionally compare against [1]–[3], which are also appropriate baselines;  we discuss their differences in detail (see **Discussion on new Related Work** section below) and include them as baselines in our experiments (see  **Updated Experiments** and **Summary of Results** sections below). **We will also add similar discussion of these baselines in the next version of the paper**
> >
> > **Discussion on new Related Work**
> >
> > **[1] and [2] both aim to detect memorization, rather than perform membership inference**. They do this by prompting the model with a fixed (50-token) prefix and scoring completions against the original text using surface-level overlap (e.g., LCS or ROUGE-L). Their goal is to surface memorized content - not to produce score distributions that cleanly separate members from non-members. This may limit their effectiveness in distinguishing borderline or weakly memorized cases. In contrast, our method aggregates signals across multiple generations using more informed n-gram overlap metrics like Creativity Index, which proves effective at distinguishing members from non-members even when memorization is subtle. Through in-depth analysis, we also identify specific design choices – such as similarity metrics, aggregation strategies, and sampling strategies – that improve performance even under weak-signal conditions.
> >
> > **[3]** introduces an attack that generates long completions (1024 tokens) and scores them using mean-aggregated ROUGE-L overlap, along with information-content normalization via zlib compression. **Our work differs in several ways:** First, we analyze multiple similarity metrics (including more “informed ones” like Creativity Index), sampling strategies, and aggregation methods, finding that these choices substantially impact performance, with optimal configurations competitive with white-box attacks. Second, we conceptually simplify the pipeline by removing "information content" normalization and leveraging only n-gram overlap statistics, showing that strong performance is achievable even without this complexity. Third, we enable dynamically constrained completions that don't exceed the original text length, significantly improving efficiency while maintaining performance. Fourth, we identify scaling strategies that systematically enhance attack effectiveness with increased compute/access. Finally, while [3] focuses on small-scale, unaligned open-source models, we evaluate on stronger contemporary testbeds including fully black-box models and reduced distribution shift settings, demonstrating robust real-world performance.
> >
> > **Updated Experiments**
> >
> > **Based on your suggestions, we run the following additional evaluation on all four datasets from the main text**:
> >
> > 1) We implement and evaluate the three additional baselines suggested.
> > * We denote the verbatim memorization attack from [1] as **VMA**.
> > * We denote the Partial Information Probing method from [2] as **PIP**.
> > * We denote the Sampling-based Pseudo-Likelihood method from [3] as **SPL** and the normalized variant **SPL (zlib)**.
> > For [1] and [2], we reimplement these methods following the descriptions in their respective papers, while we use the code directly from [3]. We use the same hyperparameters and prompts reported by the original authors for all experiments.
> > 2) Additionally, we now report both AUROC and TPR at low FPR thresholds (0.1%, 0.5%, 1.0%, and 5.0%). We focus our analysis on performance with **AUROC** and **TPR at 1% FPR**, the most commonly reported metrics in membership inference literature [5, 10-17]. Results for other FPR thresholds (0.1%, 0.5%, 5%) show similar trends and are also included in the tables below.

---

> > > ### Author Response · Authors · 2025-06-03
> > > **Response to Reviewer Yujw (Part 3)**
> > >
> > > Below, we include updated versions of all main-text tables, now expanded to include both the new baselines and the additional metrics. For clarity, tables are split up by dataset and metric. The first four columns (under “Ours →”) show our N-Gram Coverage Attack variants; the remaining columns (under “Baselines →”) correspond to the baselines. For each row, the highest-performing black-box method is **bolded**.
> > >
> > > **Summary of Results**
> > >
> > > Our updated experiments on four datasets, WikiMIA, WikiMIA Hard, TULU, and BookMIA, demonstrate strong performance across both AUROC and TPR at low FPR, providing comprehensive evidence of our method's effectiveness compared to multiple black-box baselines.
> > >
> > > First, across all datasets, our method still achieves the best or tied-best AUROC among all black-box methods. Specifically, we **outperform all baselines in 24 out of 27 model-dataset combinations** (ie, Tulu/Tulu-7B), and **tie** with exactly one other baseline in the remaining three (once with DE-COP on BookMIA, and twice with zlib-normalized SPL on WikiMIA). Again, in many cases our method approaches the performance even of white-box attacks, or sometimes even exceeds it (WikiMIA Hard, LLaMA 7B and 65B).
> > >
> > > Second, the TPR at 1% FPR corroborates the strong AUROC results in nearly all cases, demonstrating the effectiveness of N-Gram Coverage attack. On **BookMIA**, our method excels, outperforming all baselines by a margin of **at least 0.14** across all models. On **TULU**, where other black-box methods generally struggle, our method consistently achieves the highest TPR and **often approaches white-box performance**, similar to the AUROC trends. On **WikiMIA Hard**, we win convincingly on **8 out of 10 models**, while still maintaining the second-highest score on the other two models (LLaMA-30B and LLaMA-65B). On **WikiMIA**, our method leads on **2 out of 3 OpenAI models** and **1 out of 4 LLaMA models**. In the remaining cases, we lose narrowly (by **0.01** in two cases, and by 0.03 and 0.04 in the other two). Notably, on this dataset, no single baseline dominates across models – many baselines perform well on one model but poorly on others, whereas our method exhibits **consistent, competitive performance across all models**. Additionally, absolute TPR values on WikiMIA remain low for all black-box attacks (**typically near 0** and always ≤ 0.06), underscoring the difficulty of obtaining high TPR at low FPR on this benchmark. Overall, these TPR results reinforce our AUROC findings: **our method delivers strong, consistent performance across all datasets and models**, while other baselines show more erratic performance patterns.
> > >
> > > Results at other FPR thresholds (0.1%, 0.5%, 5.0%) largely corroborate the strong findings at 1.0% FPR. It's worth noting that performance at any single threshold can be somewhat arbitrary – some methods achieve early gains at extremely low FPRs but show little improvement thereafter. In contrast, our method demonstrates steady performance gains across the full range of thresholds, indicating a more balanced ROC curve while maintaining high precision throughout. This suggests our method provides strong performance regardless of the specific operating point chosen for deployment.
> > >
> > > Our results across **AUROC** and **TPR at low FPR**, evaluated against strong black-box baselines across **four diverse datasets** show that our method is **consistently effective** across model families, data domains, and evaluation metrics. Unlike the other black-box baselines (including the three newly evaluated methods: VMA, PIP, and SPL) which show inconsistent performance across these settings, our method maintains competitive results whether evaluated on books (BookMIA), recent events (WikiMIA variants), or instruction-tuning data (TULU). This robustness across diverse domains and evaluation metrics, combined with our method's simplicity and efficiency, validates our design choices and establishes N-Gram Coverage Attack as a practical and effective black-box membership inference attack.

---

> > > > ### Author Response · Authors · 2025-06-03
> > > > **Response to Reviewer Yujw (Part 4)**
> > > >
> > > > # Main Updated Results
> > > > ## BookMIA
> > > >
> > > > ### BookMIA AUROC
> > > >
> > > > | | Ours→ | | | | Baselines→ | | | | | | | | |
> > > > |---------------------|-------|----------|-------|----------|------------|------|------|------------|----------|------|--------|------|------|
> > > > | | Cov. | Cre. | LCS_C | LCS_W | PIP | VMA | SPL | SPL (zlib) | DC | Loss | R-Loss | zlib | MinK |
> > > > | GPT-3.5-0125 | 0.84 | **0.85** | 0.84 | 0.83 | 0.59 | 0.64 | 0.53 | 0.50 | 0.84 | - | - | - | - |
> > > > | GPT-3.5 Inst. | 0.91 | 0.91 | 0.92 | **0.93** | 0.73 | 0.72 | 0.46 | 0.38 | 0.68 | - | - | - | - |
> > > > | GPT-3.5-1106 | 0.84 | **0.85** | 0.83 | 0.84 | 0.57 | 0.69 | 0.51 | 0.46 | **0.85** | - | - | - | - |
> > > >
> > > > ### BookMIA TPR @ 1% FPR
> > > >
> > > > | | Ours→ | | | | Baselines→ | | | | | | | | |
> > > > |---------------------|-------|----------|-------|-------|------------|------|------|------------|------|------|--------|------|------|
> > > > | | Cov. | Cre. | LCS_C | LCS_W | PIP | VMA | SPL | SPL (zlib) | DC | Loss | R-Loss | zlib | MinK |
> > > > | GPT-3.5-0125 | 0.38 | **0.49** | 0.23 | 0.17 | 0.05 | 0.08 | 0.02 | 0.03 | 0.33 | - | - | - | - |
> > > > | GPT-3.5 Inst. | 0.61 | **0.63** | 0.50 | 0.44 | 0.27 | 0.17 | 0.01 | 0.00 | 0.16 | - | - | - | - |
> > > > | GPT-3.5-1106 | 0.52 | **0.58** | 0.34 | 0.23 | 0.06 | 0.10 | 0.02 | 0.01 | 0.44 | - | - | - | - |
> > > >
> > > > ## Tulu
> > > > ### Tulu AUROC
> > > > | | Ours→ | | | | Baselines→ | | | | | | | | |
> > > > |--------------|----------|----------|-------|-------|------------|------|------|------------|--------|------|--------|------|------|
> > > > | | Cov. | Cre. | LCS_C | LCS_W | PIP | VMA | SPL | SPL (zlib) | DC | Loss | R-Loss | zlib | MinK |
> > > > | Tulu-7B | **0.79** | **0.79** | 0.73 | 0.74 | 0.50 | 0.59 | 0.49 | 0.44 | 0.48 | 0.84 | 0.50 | 0.81 | 0.84 |
> > > > | Tulu-13B | **0.80** | **0.80** | 0.74 | 0.76 | 0.50 | 0.61 | 0.49 | 0.44 | 0.47 | 0.87 | 0.63 | 0.83 | 0.87 |
> > > > | Tulu-30B | **0.82** | **0.82** | 0.76 | 0.77 | 0.51 | 0.61 | 0.49 | 0.44 | 0.44 | 0.87 | 0.54 | 0.84 | 0.87 |
> > > > | Tulu-65B | 0.85 | **0.86** | 0.80 | 0.80 | 0.54 | 0.64 | 0.51 | 0.44 | 0.44 | 0.92 | 0.68 | 0.90 | 0.92 |
> > > > | Tulu-1.1-7B | 0.72 | **0.73** | 0.70 | 0.71 | 0.52 | 0.59 | 0.46 | 0.42 | 0.4328 | 0.77 | 0.50 | 0.74 | 0.76 |
> > > > | Tulu-1.1-13B | **0.76** | 0.75 | 0.71 | 0.72 | 0.50 | 0.58 | 0.47 | 0.42 | 0.4542 | 0.81 | 0.58 | 0.77 | 0.81 |
> > > > | Tulu-1.1-70B | **0.79** | 0.78 | 0.75 | 0.77 | 0.50 | 0.60 | 0.46 | 0.40 | 0.4669 | 0.86 | 0.64 | 0.84 | 0.86 |
> > > >
> > > > ### Tulu TPR @ 1% FPR
> > > >
> > > > | | Ours→ | | | | Baselines→ | | | | | | | | |
> > > > |--------------|-------|----------|----------|----------|------------|------|------|------------|------|------|--------|------|------|
> > > > | | Cov. | Cre. | LCS_C | LCS_W | PIP | VMA | SPL | SPL (zlib) | DC | Loss | R-Loss | zlib | MinK |
> > > > | Tulu-7B | 0.00 | 0.00 | **0.12** | 0.08 | 0.04 | 0.02 | 0.01 | 0.00 | 0.00 | 0.08 | 0.00 | 0.25 | 0.08 |
> > > > | Tulu-13B | 0.00 | 0.16 | **0.17** | 0.14 | 0.03 | 0.05 | 0.01 | 0.00 | 0.00 | 0.13 | 0.02 | 0.28 | 0.13 |
> > > > | Tulu-30B | 0.00 | 0.00 | **0.14** | 0.13 | 0.02 | 0.05 | 0.01 | 0.01 | 0.00 | 0.13 | 0.00 | 0.29 | 0.12 |
> > > > | Tulu-65B | 0.00 | **0.23** | **0.23** | 0.22 | 0.06 | 0.07 | 0.02 | 0.00 | 0.00 | 0.24 | 0.01 | 0.44 | 0.24 |
> > > > | Tulu-1.1-7B | 0.00 | 0.07 | **0.10** | 0.08 | 0.03 | 0.04 | 0.01 | 0.00 | 0.00 | 0.07 | 0.00 | 0.19 | 0.07 |
> > > > | Tulu-1.1-13B | 0.00 | 0.08 | **0.11** | **0.11** | 0.02 | 0.04 | 0.01 | 0.00 | 0.00 | 0.12 | 0.01 | 0.24 | 0.12 |
> > > > | Tulu-1.1-70B | 0.00 | 0.12 | 0.18 | **0.21** | 0.02 | 0.05 | 0.00 | 0.00 | 0.00 | 0.07 | 0.02 | 0.12 | 0.08 |
> > > >
> > > > ## WikiMIA_2024 Hard
> > > > ### Wikimia_2024 Hard AUROC
> > > > | | Ours→ | | | | Baselines→ | | | | | | | | |
> > > > |---------------|----------|----------|-------|-------|------------|------|------|------------|------|------|--------|------|------|
> > > > | | Cov. | Cre. | LCS_C | LCS_W | PIP | VMA | SPL | SPL (zlib) | DC | Loss | R-Loss | zlib | MinK |
> > > > | GPT-3.5-0125 | **0.59** | 0.57 | 0.55 | 0.55 | 0.50 | 0.50 | 0.46 | 0.47 | 0.47 | - | - | - | - |
> > > > | GPT-3.5 Inst. | **0.64** | 0.63 | 0.61 | 0.61 | 0.56 | 0.50 | 0.48 | 0.49 | 0.45 | - | - | - | - |
> > > > | GPT-3.5-1106 | **0.58** | **0.58** | 0.56 | 0.57 | 0.46 | 0.51 | 0.46 | 0.46 | 0.49 | - | - | - | - |
> > > > | GPT-4 | 0.57 | **0.58** | 0.55 | 0.57 | 0.51 | 0.50 | 0.46 | 0.46 | 0.44 | - | - | - | - |
> > > > | GPT-4o-1120 | **0.55** | **0.55** | 0.54 | 0.52 | 0.51 | 0.50 | 0.44 | 0.44 | 0.51 | - | - | - | - |
> > > > | GPT-4o Mini | **0.55** | 0.53 | 0.52 | 0.52 | 0.51 | 0.50 | 0.46 | 0.46 | 0.43 | - | - | - | - |
> > > > | LLaMA-7B | **0.55** | 0.54 | 0.57 | 0.54 | 0.54 | 0.51 | 0.53 | 0.51 | 0.47 | - | - | 0.50 | 0.52 |
> > > > | LLaMA-13B | **0.59** | 0.58 | 0.57 | 0.54 | 0.54 | 0.51 | 0.47 | 0.47 | 0.51 | 0.53 | 0.57 | 0.51 | 0.54 |
> > > > | LLaMA-30B | **0.61** | **0.61** | 0.60 | 0.57 | 0.58 | 0.52 | 0.49 | 0.51 | 0.50 | 0.56 | 0.61 | 0.53 | 0.60 |
> > > > | LLaMA-65B | **0.64** | 0.63 | 0.60 | 0.60 | 0.58 | 0.53 | 0.50 | 0.47 | 0.51 | 0.57 | 0.61 | 0.54 | 0.58 |

---

> > > > > ### Author Response · Authors · 2025-06-03
> > > > > **Response to Reviewer Yujw (Part 5)**
> > > > >
> > > > > ### Wikimia_2024 Hard TPR @ 1% FPR
> > > > > | | Ours→ | | | | Baselines→ | | | | | | | | |
> > > > > |---------------|----------|----------|----------|----------|------------|----------|------|------------|------|------|--------|------|------|
> > > > > | | Cov. | Cre. | LCS_C | LCS_W | PIP | VMA | SPL | SPL (zlib) | DC | Loss | R-Loss | zlib | MinK |
> > > > > | GPT-3.5-0125 | 0.02 | 0.01 | 0.03 | **0.04** | 0.01 | 0.04 | 0.01 | 0.01 | 0.02 | - | - | - | - |
> > > > > | GPT-3.5 Inst. | 0.05 | **0.06** | 0.04 | **0.06** | **0.06** | 0.01 | 0.01 | 0.01 | 0.01 | - | - | - | - |
> > > > > | GPT-3.5-1106 | 0.04 | **0.08** | 0.07 | 0.06 | 0.02 | 0.04 | 0.01 | 0.01 | 0.03 | - | - | - | - |
> > > > > | GPT-4 | 0.01 | 0.04 | 0.05 | **0.08** | 0.04 | 0.02 | 0.00 | 0.00 | 0.00 | - | - | - | - |
> > > > > | GPT-4o-1120 | **0.03** | 0.02 | 0.02 | **0.03** | 0.02 | 0.01 | 0.01 | 0.00 | 0.00 | - | - | - | - |
> > > > > | GPT-4o Mini | **0.04** | 0.03 | 0.02 | 0.01 | 0.02 | 0.02 | 0.01 | 0.01 | 0.00 | - | - | - | - |
> > > > > | LLaMA-7B | 0.04 | 0.04 | **0.05** | 0.01 | 0.04 | 0.04 | 0.00 | 0.04 | 0.03 | 0.03 | - | 0.02 | 0.04 |
> > > > > | LLaMA-13B | 0.00 | 0.02 | 0.04 | 0.02 | **0.12** | 0.03 | 0.02 | 0.02 | 0.05 | 0.04 | 0.05 | 0.04 | 0.03 |
> > > > > | LLaMA-30B | 0.00 | 0.05 | 0.09 | 0.04 | 0.00 | **0.12** | 0.04 | 0.04 | 0.00 | 0.16 | 0.07 | 0.11 | 0.09 |
> > > > > | LLaMA-65B | 0.00 | 0.09 | 0.08 | 0.10 | **0.16** | 0.08 | 0.00 | 0.00 | 0.00 | 0.18 | 0.09 | 0.13 | 0.10 |
> > > > >
> > > > > ## WikiMIA
> > > > > ### Wikimia AUROC
> > > > > | | Ours→ | | | | Baselines→ | | | | | | | | |
> > > > > |---------------|----------|------|-------|-------|------------|------|------|------------|------|------|--------|------|------|
> > > > > | | Cov. | Cre. | LCS_C | LCS_W | PIP | VMA | SPL | SPL (zlib) | DC | Loss | R-Loss | zlib | MinK |
> > > > > | GPT-3.5-0125 | **0.64** | 0.63 | 0.61 | 0.60 | 0.59 | 0.58 | 0.57 | 0.54 | 0.54 | - | - | - | - |
> > > > > | GPT-3.5 Inst. | **0.62** | 0.61 | 0.58 | 0.58 | 0.61 | 0.57 | 0.58 | 0.57 | 0.54 | - | - | - | - |
> > > > > | GPT-3.5-1106 | **0.64** | 0.62 | 0.61 | 0.60 | 0.62 | 0.61 | 0.55 | 0.50 | 0.52 | - | - | - | - |
> > > > > | LLaMA-7B | **0.60** | 0.59 | 0.56 | 0.55 | 0.54 | 0.55 | 0.58 | **0.60** | 0.48 | 0.62 | - | 0.63 | 0.64 |
> > > > > | LLaMA-13B | **0.62** | 0.59 | 0.57 | 0.54 | 0.58 | 0.55 | 0.61 | **0.62** | 0.52 | 0.64 | 0.63 | 0.65 | 0.66 |
> > > > > | LLaMA-30B | **0.63** | 0.62 | 0.57 | 0.58 | 0.59 | 0.57 | 0.56 | 0.56 | 0.49 | 0.66 | 0.69 | 0.67 | 0.69 |
> > > > > | LLaMA-65B | **0.65** | 0.64 | 0.61 | 0.58 | 0.60 | 0.59 | 0.62 | 0.60 | 0.50 | 0.68 | 0.74 | 0.69 | 0.70 |
> > > > >
> > > > > ### Wikimia TPR @ 1% FPR
> > > > >
> > > > > | | Ours→ | | | | Baselines→ | | | | | | | | |
> > > > > |---------------|----------|----------|-------|-------|------------|------|------|------------|----------|------|--------|------|------|
> > > > > | | Cov. | Cre. | LCS_C | LCS_W | PIP | VMA | SPL | SPL (zlib) | DC | Loss | R-Loss | zlib | MinK |
> > > > > | GPT-3.5-0125 | **0.05** | 0.02 | 0.02 | 0.01 | 0.01 | 0.00 | 0.00 | 0.00 | **0.05** | - | - | - | - |
> > > > > | GPT-3.5 Inst. | 0.02 | **0.03** | 0.01 | 0.01 | 0.00 | 0.00 | 0.00 | 0.00 | 0.02 | - | - | - | - |
> > > > > | GPT-3.5-1106 | 0.04 | 0.03 | 0.02 | 0.02 | 0.03 | 0.01 | 0.00 | 0.00 | **0.05** | - | - | - | - |
> > > > > | LLaMA-7B | 0.02 | 0.02 | 0.02 | 0.02 | 0.03 | 0.02 | 0.02 | **0.06** | 0.00 | 0.06 | - | 0.08 | 0.08 |
> > > > > | LLaMA-13B | 0.01 | 0.01 | 0.00 | 0.02 | 0.02 | 0.03 | 0.04 | **0.05** | 0.04 | 0.08 | 0.03 | 0.09 | 0.08 |
> > > > > | LLaMA-30B | 0.02 | 0.01 | 0.04 | 0.01 | 0.00 | 0.04 | 0.01 | 0.01 | **0.05** | 0.07 | 0.02 | 0.10 | 0.08 |
> > > > > | LLaMA-65B | 0.01 | 0.03 | 0.02 | 0.03 | **0.06** | 0.05 | 0.03 | 0.02 | 0.01 | 0.10 | 0.00 | 0.14 | 0.13 |
> > > > >
> > > > > # Additional Updated Results (TPR @ other FPR thresholds)
> > > > >
> > > > > ## BookMIA
> > > > > ### bookMIA TPR @ 0.1% FPR
> > > > > | | Ours→ | | | | Baselines→ | | | | | | | | |
> > > > > |---------------------|-------|----------|-------|-------|------------|------|------|------------|------|------|--------|------|------|
> > > > > | | Cov. | Cre. | LCS_C | LCS_W | PIP | VMA | SPL | SPL (zlib) | DC | Loss | R-Loss | zlib | MinK |
> > > > > | GPT-3.5-0125 | 0.35 | **0.42** | 0.19 | 0.08 | 0.03 | 0.05 | 0.01 | 0.01 | 0.23 | - | - | - | - |
> > > > > | GPT-3.5 Inst. | 0.51 | **0.52** | 0.38 | 0.42 | 0.20 | 0.14 | 0.01 | 0.00 | 0.04 | - | - | - | - |
> > > > > | GPT-3.5-1106 | 0.35 | **0.39** | 0.16 | 0.05 | 0.04 | 0.08 | 0.01 | 0.00 | 0.18 | - | - | - | - |
> > > > > ### bookMIA TPR @ 0.5% FPR
> > > > > | | Ours→ | | | | Baselines→ | | | | | | | | |
> > > > > |---------------------|-------|----------|-------|-------|------------|------|------|------------|------|------|--------|------|------|
> > > > > | | Cov. | Cre. | LCS_C | LCS_W | PIP | VMA | SPL | SPL (zlib) | DC | Loss | R-Loss | zlib | MinK |
> > > > > | GPT-3.5-0125 | 0.37 | **0.47** | 0.19 | 0.14 | 0.04 | 0.05 | 0.02 | 0.03 | 0.27 | - | - | - | - |
> > > > > | GPT-3.5 Inst. | 0.56 | **0.59** | 0.50 | 0.44 | 0.25 | 0.14 | 0.01 | 0.00 | 0.08 | - | - | - | - |
> > > > > | GPT-3.5-1106 | 0.51 | **0.58** | 0.30 | 0.08 | 0.05 | 0.08 | 0.02 | 0.00 | 0.37 | - | - | - | - |

---

> > > > > > ### Author Response · Authors · 2025-06-03
> > > > > > **Response to Reviewer Yujw (Part 6)**
> > > > > >
> > > > > > ### bookMIA TPR @ 5% FPR
> > > > > > | | Ours→ | | | | Baselines→ | | | | | | | | |
> > > > > > |---------------------|-------|----------|-------|-------|------------|------|------|------------|------|------|--------|------|------|
> > > > > > | | Cov. | Cre. | LCS_C | LCS_W | PIP | VMA | SPL | SPL (zlib) | DC | Loss | R-Loss | zlib | MinK |
> > > > > > | GPT-3.5-0125 | 0.52 | **0.59** | 0.52 | 0.48 | 0.16 | 0.17 | 0.08 | 0.07 | 0.54 | 0.40 | 0.00 | 0.43 | 0.41 |
> > > > > > | GPT-3.5 Inst. | 0.75 | **0.77** | 0.66 | 0.71 | 0.34 | 0.31 | 0.03 | 0.01 | 0.31 | 0.55 | 0.06 | 0.49 | 0.55 |
> > > > > > | GPT-3.5-1106 | 0.59 | **0.62** | 0.55 | 0.53 | 0.12 | 0.21 | 0.07 | 0.07 | 0.55 | 0.45 | 0.03 | 0.50 | 0.45 |
> > > > > > ## Tulu
> > > > > > ### Tulu TPR @ 0.1% FPR
> > > > > > |              | Ours→ |      |       |       | Baselines→ |          |      |            |      |      |        |      |      |
> > > > > > |--------------|-------|------|-------|-------|------------|----------|------|------------|------|------|--------|------|------|
> > > > > > |              | Cov.  | Cre. | LCS_C | LCS_W | PIP        | VMA      | SPL  | SPL (zlib) | DC   | Loss | R-Loss | zlib | MinK |
> > > > > > | Tulu-7B      | 0.00  | 0.00 | 0.00  | 0.00  | 0.00       | 0.00     | 0.00 | 0.00       | 0.00 | 0.00 | 0.00   | 0.00 | 0.00 |
> > > > > > | Tulu-13B     | 0.00  | 0.00 | 0.00  | 0.00  | 0.01       | **0.02** | 0.00 | 0.00       | 0.00 | 0.00 | 0.01   | 0.01 | 0.00 |
> > > > > > | Tulu-30B     | 0.00  | 0.00 | 0.00  | 0.00  | 0.00       | **0.02** | 0.00 | 0.00       | 0.00 | 0.01 | 0.00   | 0.01 | 0.01 |
> > > > > > | Tulu-65B     | 0.00  | 0.00 | 0.00  | 0.00  | 0.01       | **0.03** | 0.00 | 0.00       | 0.00 | 0.04 | 0.00   | 0.01 | 0.04 |
> > > > > > | Tulu-1.1-7B  | 0.00  | 0.00 | 0.00  | 0.00  | **0.01**   | **0.01** | 0.00 | 0.00       | 0.00 | 0.00 | 0.00   | 0.00 | 0.00 |
> > > > > > | Tulu-1.1-13B | 0.00  | 0.00 | 0.00  | 0.00  | 0.00       | **0.01** | 0.00 | 0.00       | 0.00 | 0.01 | 0.00   | 0.00 | 0.01 |
> > > > > > | Tulu-1.1-70B | 0.00  | 0.00 | 0.00  | 0.00  | 0.00       | **0.02** | 0.00 | 0.00       | 0.00 | 0.01 | 0.00   | 0.03 | 0.01 |
> > > > > > ### Tulu TPR @ 0.5% FPR
> > > > > > | | Ours→ | | | | Baselines→ | | | | | | | | |
> > > > > > |--------------|-------|------|----------|----------|------------|------|------|------------|------|------|--------|------|------|
> > > > > > | | Cov. | Cre. | LCS_C | LCS_W | PIP | VMA | SPL | SPL (zlib) | DC | Loss | R-Loss | zlib | MinK |
> > > > > > | Tulu-7B | 0.00 | 0.00 | 0.01 | **0.03** | **0.03** | 0.02 | 0.00 | 0.00 | 0.00 | 0.04 | 0.00 | 0.04 | 0.04 |
> > > > > > | Tulu-13B | 0.00 | 0.00 | **0.12** | 0.05 | 0.02 | 0.02 | 0.00 | 0.00 | 0.00 | 0.04 | 0.02 | 0.06 | 0.04 |
> > > > > > | Tulu-30B | 0.00 | 0.00 | **0.11** | 0.08 | 0.00 | 0.03 | 0.01 | 0.00 | 0.00 | 0.03 | 0.00 | 0.09 | 0.03 |
> > > > > > | Tulu-65B | 0.00 | 0.00 | **0.12** | 0.10 | 0.05 | 0.06 | 0.01 | 0.00 | 0.00 | 0.09 | 0.00 | 0.22 | 0.09 |
> > > > > > | Tulu-1.1-7B | 0.00 | 0.00 | **0.06** | 0.04 | 0.02 | 0.02 | 0.01 | 0.00 | 0 | 0.05 | 0.00 | 0.10 | 0.05 |
> > > > > > | Tulu-1.1-13B | 0.00 | 0.00 | **0.05** | 0.05 | 0.00 | 0.04 | 0.00 | 0.00 | 0 | 0.03 | 0.00 | 0.18 | 0.03 |
> > > > > > | Tulu-1.1-70B | 0.00 | 0.00 | **0.18** | 0.16 | 0.00 | 0.04 | 0.00 | 0.00 | 0 | 0.02 | 0.00 | 0.06 | 0.02 |
> > > > > > ### Tulu TPR @ 5% FPR
> > > > > > | | Ours→ | | | | Baselines→ | | | | | | | | |
> > > > > > |--------------|-------|------|----------|----------|------------|------|------|------------|------|------|--------|------|------|
> > > > > > | | Cov. | Cre. | LCS_C | LCS_W | PIP | VMA | SPL | SPL (zlib) | DC | Loss | R-Loss | zlib | MinK |
> > > > > > | Tulu-7B | 0.00 | 0.27 | **0.35** | 0.32 | 0.11 | 0.10 | 0.03 | 0.02 | 0.03 | 0.40 | 0.00 | 0.43 | 0.41 |
> > > > > > | Tulu-13B | 0.00 | 0.32 | 0.35 | **0.38** | 0.10 | 0.11 | 0.04 | 0.02 | 0.03 | 0.55 | 0.06 | 0.49 | 0.55 |
> > > > > > | Tulu-30B | 0.00 | 0.35 | **0.39** | **0.39** | 0.05 | 0.14 | 0.04 | 0.01 | 0.00 | 0.45 | 0.03 | 0.50 | 0.45 |
> > > > > > | Tulu-65B | 0.00 | 0.43 | 0.46 | **0.47** | 0.13 | 0.17 | 0.06 | 0.01 | 0.00 | 0.67 | 0.09 | 0.70 | 0.67 |
> > > > > > | Tulu-1.1-7B | 0.00 | 0.22 | 0.29 | **0.31** | 0.08 | 0.11 | 0.03 | 0.02 | 0 | 0.36 | 0.00 | 0.36 | 0.36 |
> > > > > > | Tulu-1.1-13B | 0.00 | 0.27 | 0.31 | **0.35** | 0.05 | 0.12 | 0.03 | 0.02 | 0 | 0.44 | 0.08 | 0.41 | 0.44 |
> > > > > > | Tulu-1.1-70B | 0.00 | 0.33 | 0.37 | **0.40** | 0.06 | 0.14 | 0.03 | 0.02 | 0.04 | 0.43 | 0.12 | 0.54 | 0.44 |

---

> > > > > > > ### Author Response · Authors · 2025-06-03
> > > > > > > **Response to Reviewer Yujw (Part 7)**
> > > > > > >
> > > > > > > ## WikiMIA_2024 Hard
> > > > > > > ### Wikimia_2024 Hard TPR @ 0.1% FPR
> > > > > > >
> > > > > > > | | Ours→ | | | | Baselines→ | | | | | | | | |
> > > > > > > |---------------|----------|----------|----------|----------|------------|----------|------|------------|------|------|--------|------|------|
> > > > > > > | | Cov. | Cre. | LCS_C | LCS_W | PIP | VMA | SPL | SPL (zlib) | DC | Loss | R-Loss | zlib | MinK |
> > > > > > > | GPT-3.5-0125 | 0.02 | 0.01 | 0.03 | 0.00 | 0.01 | **0.04** | 0.01 | 0.00 | 0.00 | - | - | - | - |
> > > > > > > | GPT-3.5 Inst. | 0.03 | **0.05** | 0.02 | 0.01 | 0.00 | 0.01 | 0.01 | 0.01 | 0.01 | - | - | - | - |
> > > > > > > | GPT-3.5-1106 | 0.02 | 0.02 | 0.05 | **0.06** | 0.01 | 0.04 | 0.01 | 0.00 | 0.03 | - | - | - | - |
> > > > > > > | GPT-4 | 0.01 | 0.02 | **0.04** | **0.04** | **0.04** | 0.01 | 0.00 | 0.00 | 0.00 | - | - | - | - |
> > > > > > > | GPT-4o-1120 | **0.02** | **0.02** | 0.01 | 0.01 | 0.01 | 0.01 | 0.00 | 0.00 | 0.00 | - | - | - | - |
> > > > > > > | GPT-4o Mini | **0.04** | 0.02 | 0.01 | 0.01 | 0.01 | 0.02 | 0.00 | 0.00 | 0.00 | - | - | - | - |
> > > > > > > | LLaMA-7B | 0.00 | 0.02 | **0.05** | 0.01 | 0.02 | 0.01 | 0.00 | 0.02 | 0.00 | 0.03 | - | 0.00 | 0.01 |
> > > > > > > | LLaMA-13B | 0.00 | 0.01 | 0.02 | 0.02 | **0.12** | 0.03 | 0.02 | 0.00 | 0.01 | 0.03 | 0.04 | 0.03 | 0.03 |
> > > > > > > | LLaMA-30B | 0.00 | 0.05 | 0.05 | 0.04 | 0.00 | **0.09** | 0.03 | 0.04 | 0.00 | 0.14 | 0.03 | 0.10 | 0.04 |
> > > > > > > | LLaMA-65B | 0.00 | 0.08 | 0.07 | **0.10** | 0.07 | 0.04 | 0.00 | 0.00 | 0.00 | 0.16 | 0.04 | 0.12 | 0.09 |
> > > > > > > ### Wikimia_2024 Hard TPR @ 0.5% FPR
> > > > > > > | | Ours→ | | | | Baselines→ | | | | | | | | |
> > > > > > > |---------------|----------|----------|----------|----------|------------|----------|------|------------|------|------|--------|------|------|
> > > > > > > | | Cov. | Cre. | LCS_C | LCS_W | PIP | VMA | SPL | SPL (zlib) | DC | Loss | R-Loss | zlib | MinK |
> > > > > > > | GPT-3.5-0125 | 0.02 | 0.01 | 0.03 | 0.00 | 0.01 | **0.04** | 0.01 | 0.00 | 0.00 | - | - | - | - |
> > > > > > > | GPT-3.5 Inst. | 0.03 | **0.05** | 0.02 | 0.01 | 0.00 | 0.01 | 0.01 | 0.01 | 0.01 | - | - | - | - |
> > > > > > > | GPT-3.5-1106 | 0.02 | 0.02 | 0.05 | **0.06** | 0.01 | 0.04 | 0.01 | 0.00 | 0.03 | - | - | - | - |
> > > > > > > | GPT-4 | 0.01 | 0.02 | **0.04** | **0.04** | **0.04** | 0.01 | 0.00 | 0.00 | 0.00 | - | - | - | - |
> > > > > > > | GPT-4o-1120 | **0.02** | **0.02** | 0.01 | 0.01 | 0.01 | 0.01 | 0.00 | 0.00 | 0.00 | - | - | - | - |
> > > > > > > | GPT-4o Mini | **0.04** | 0.02 | 0.01 | 0.01 | 0.01 | 0.02 | 0.00 | 0.00 | 0.00 | - | - | - | - |
> > > > > > > | LLaMA-7B | 0.00 | 0.02 | **0.05** | 0.01 | 0.02 | 0.01 | 0.00 | 0.02 | 0.00 | 0.03 | - | 0.00 | 0.01 |
> > > > > > > | LLaMA-13B | 0.00 | 0.01 | 0.02 | 0.02 | **0.12** | 0.03 | 0.02 | 0.00 | 0.01 | 0.03 | 0.04 | 0.03 | 0.03 |
> > > > > > > | LLaMA-30B | 0.00 | 0.05 | 0.05 | 0.04 | 0.00 | **0.09** | 0.03 | 0.04 | 0.00 | 0.14 | 0.03 | 0.10 | 0.04 |
> > > > > > > | LLaMA-65B | 0.00 | 0.08 | 0.07 | **0.10** | 0.07 | 0.04 | 0.00 | 0.00 | 0.00 | 0.16 | 0.04 | 0.12 | 0.09 |
> > > > > > > ### Wikimia_2024 Hard TPR @ 5% FPR
> > > > > > > | | Ours→ | | | | Baselines→ | | | | | | | | |
> > > > > > > |---------------|----------|----------|----------|----------|------------|------|------|------------|------|------|--------|------|------|
> > > > > > > | | Cov. | Cre. | LCS_C | LCS_W | PIP | VMA | SPL | SPL (zlib) | DC | Loss | R-Loss | zlib | MinK |
> > > > > > > | GPT-3.5-0125 | **0.10** | 0.09 | 0.06 | 0.09 | 0.06 | 0.06 | 0.03 | 0.02 | 0.06 | - | - | - | - |
> > > > > > > | GPT-3.5 Inst. | 0.12 | 0.14 | 0.13 | **0.15** | **0.15** | 0.07 | 0.03 | 0.03 | 0.01 | - | - | - | - |
> > > > > > > | GPT-3.5-1106 | 0.09 | 0.11 | 0.10 | **0.15** | 0.09 | 0.10 | 0.05 | 0.04 | 0.04 | - | - | - | - |
> > > > > > > | GPT-4 | 0.10 | 0.05 | **0.13** | 0.11 | 0.10 | 0.06 | 0.04 | 0.06 | 0.00 | - | - | - | - |
> > > > > > > | GPT-4o-1120 | 0.09 | **0.13** | 0.10 | 0.06 | 0.08 | 0.08 | 0.02 | 0.03 | 0.00 | - | - | - | - |
> > > > > > > | GPT-4o Mini | **0.08** | 0.07 | 0.02 | 0.06 | 0.04 | 0.05 | 0.02 | 0.04 | 0.04 | - | - | - | - |
> > > > > > > | LLaMA-7B | 0.10 | **0.17** | 0.06 | 0.03 | 0.12 | 0.04 | 0.04 | 0.04 | 0.03 | 0.08 | - | 0.07 | 0.09 |
> > > > > > > | LLaMA-13B | 0.00 | **0.15** | 0.08 | 0.05 | **0.15** | 0.08 | 0.03 | 0.03 | 0.06 | 0.10 | 0.11 | 0.09 | 0.06 |
> > > > > > > | LLaMA-30B | 0.00 | 0.09 | 0.13 | 0.13 | **0.16** | 0.13 | 0.04 | 0.05 | 0.05 | 0.20 | 0.17 | 0.13 | 0.16 |
> > > > > > > | LLaMA-65B | **0.24** | 0.14 | 0.13 | 0.13 | 0.20 | 0.09 | 0.04 | 0.02 | 0.02 | 0.20 | 0.21 | 0.14 | 0.18 |

---

> > > > > > > > ### Author Response · Authors · 2025-06-03
> > > > > > > > **Response to Reviewer Yujw (Part 8)**
> > > > > > > >
> > > > > > > > ## WikiMIA
> > > > > > > > ### WikiMIA TPR @ 0.1% FPR
> > > > > > > > | | Ours→ | | | | Baselines→ | | | | | | | | |
> > > > > > > > |---------------|----------|----------|-------|----------|------------|----------|----------|------------|------|------|--------|------|------|
> > > > > > > > | | Cov. | Cre. | LCS_C | LCS_W | PIP | VMA | SPL | SPL (zlib) | DC | Loss | R-Loss | zlib | MinK |
> > > > > > > > | GPT-3.5-0125 | 0.00 | 0.00 | 0.00 | 0.00 | 0.00 | 0.00 | 0.00 | 0.00 | 0.03 | - | - | - | - |
> > > > > > > > | GPT-3.5 Inst. | 0.00 | **0.02** | 0.00 | 0.01 | 0.00 | 0.00 | 0.00 | 0.00 | 0.00 | - | - | - | - |
> > > > > > > > | GPT-3.5-1106 | 0.00 | 0.00 | 0.00 | **0.02** | 0.00 | 0.00 | 0.00 | 0.00 | 0.02 | - | - | - | - |
> > > > > > > > | LLaMA-7B | **0.02** | 0.00 | 0.00 | 0.00 | 0.01 | **0.02** | 0.00 | **0.02** | 0.00 | 0.04 | - | 0.04 | 0.04 |
> > > > > > > > | LLaMA-13B | 0.00 | 0.00 | 0.00 | 0.00 | 0.00 | 0.02 | 0.03 | **0.04** | 0.01 | 0.04 | 0.00 | 0.07 | 0.05 |
> > > > > > > > | LLaMA-30B | 0.00 | 0.00 | 0.01 | 0.00 | 0.00 | **0.02** | 0.01 | 0.00 | 0.00 | 0.03 | 0.00 | 0.04 | 0.04 |
> > > > > > > > | LLaMA-65B | 0.00 | **0.02** | 0.01 | **0.02** | 0.00 | **0.02** | **0.02** | 0.01 | 0.01 | 0.03 | 0.00 | 0.03 | 0.03 |
> > > > > > > >
> > > > > > > > ### Wikimia TPR @ 0.5%FPR
> > > > > > > > | | Ours→ | | | | Baselines→ | | | | | | | | |
> > > > > > > > |---------------|----------|----------|-------|-------|------------|------|------|------------|----------|------|--------|------|------|
> > > > > > > > | | Cov. | Cre. | LCS_C | LCS_W | PIP | VMA | SPL | SPL (zlib) | DC | Loss | R-Loss | zlib | MinK |
> > > > > > > > | GPT-3.5-0125 | **0.04** | 0.00 | 0.00 | 0.01 | 0.01 | 0.00 | 0.00 | 0.00 | 0.03 | - | - | - | - |
> > > > > > > > | GPT-3.5 Inst. | **0.02** | **0.02** | 0.01 | 0.01 | 0.00 | 0.00 | 0.00 | 0.00 | 0.00 | - | - | - | - |
> > > > > > > > | GPT-3.5-1106 | 0.01 | 0.02 | 0.02 | 0.02 | 0.00 | 0.01 | 0.00 | 0.00 | **0.03** | - | - | - | - |
> > > > > > > > | LLaMA-7B | 0.02 | 0.00 | 0.00 | 0.00 | 0.02 | 0.02 | 0.02 | **0.05** | 0.00 | 0.04 | - | 0.06 | 0.05 |
> > > > > > > > | LLaMA-13B | 0.01 | 0.01 | 0.00 | 0.01 | 0.02 | 0.03 | 0.04 | **0.04** | **0.04** | 0.05 | 0.01 | 0.09 | 0.06 |
> > > > > > > > | LLaMA-30B | 0.00 | 0.01 | 0.03 | 0.01 | 0.00 | 0.02 | 0.01 | 0.00 | **0.05** | 0.06 | 0.00 | 0.10 | 0.08 |
> > > > > > > > | LLaMA-65B | 0.00 | 0.03 | 0.01 | 0.03 | **0.05** | 0.02 | 0.03 | 0.02 | 0.01 | 0.10 | 0.00 | 0.13 | 0.12 |
> > > > > > > > ### Wikimia TPR @ 5% FPR
> > > > > > > > | | Ours→ | | | | Baselines→ | | | | | | | | |
> > > > > > > > |---------------|----------|----------|-------|-------|------------|------|------|------------|----------|------|--------|------|------|
> > > > > > > > | | Cov. | Cre. | LCS_C | LCS_W | PIP | VMA | SPL | SPL (zlib) | DC | Loss | R-Loss | zlib | MinK |
> > > > > > > > | GPT-3.5-0125 | 0.11 | 0.10 | 0.06 | 0.07 | 0.06 | 0.08 | 0.04 | 0.03 | **0.15** | - | - | - | - |
> > > > > > > > | GPT-3.5 Inst. | 0.11 | 0.09 | 0.07 | 0.08 | **0.12** | 0.07 | 0.03 | 0.03 | 0.07 | - | - | - | - |
> > > > > > > > | GPT-3.5-1106 | **0.13** | **0.13** | 0.08 | 0.10 | 0.11 | 0.07 | 0.03 | 0.03 | 0.11 | - | - | - | - |
> > > > > > > > | LLaMA-7B | 0.10 | 0.08 | 0.06 | 0.05 | 0.09 | 0.04 | 0.06 | **0.12** | 0.02 | 0.08 | - | 0.12 | 0.17 |
> > > > > > > > | LLaMA-13B | 0.08 | 0.05 | 0.05 | 0.04 | **0.13** | 0.07 | 0.06 | 0.10 | 0.06 | 0.13 | 0.04 | 0.14 | 0.18 |
> > > > > > > > | LLaMA-30B | 0.08 | **0.11** | 0.05 | 0.10 | 0.08 | 0.06 | 0.10 | 0.10 | 0.10 | 0.15 | 0.11 | 0.17 | 0.18 |
> > > > > > > > | LLaMA-65B | 0.10 | **0.11** | 0.09 | 0.07 | 0.10 | 0.10 | 0.09 | 0.05 | 0.01 | 0.17 | 0.14 | 0.18 | 0.20 |

---

> > > > > > > > > ### Author Response · Authors · 2025-06-03
> > > > > > > > > **Response to Reviewer Yujw (Part 9)**
> > > > > > > > >
> > > > > > > > > # References
> > > > > > > > > [1] Karamolegkou, A., Li, J., Zhou, L., & Søgaard, A. (2023). Copyright Violations and Large Language Models. ArXiv, abs/2310.13771.
> > > > > > > > >
> > > > > > > > > [2] Zhao, W., Shao, H., Xu, Z., Duan, S., & Zhang, D. (2024). Measuring Copyright Risks of Large Language Model via Partial Information Probing. ArXiv, abs/2409.13831.
> > > > > > > > >
> > > > > > > > > [3] Kaneko, M., Ma, Y., Wata, Y., & Okazaki, N. (2024). Sampling-based Pseudo-Likelihood for Membership Inference Attacks. ArXiv, abs/2404.11262.
> > > > > > > > >
> > > > > > > > > [4] Das, D., Zhang, J., & Tramèr, F. (2024). Blind Baselines Beat Membership Inference Attacks for Foundation Models. ArXiv, abs/2406.16201.
> > > > > > > > >
> > > > > > > > > [5] Duan, M., Suri, A., Mireshghallah, N., Min, S., Shi, W., Zettlemoyer, L.S., Tsvetkov, Y., Choi, Y., Evans, D., & Hajishirzi, H. (2024). Do Membership Inference Attacks Work on Large Language Models? ArXiv, abs/2402.07841.
> > > > > > > > >
> > > > > > > > > [6] Duarte, A.V., Zhao, X., Oliveira, A.L., & Li, L. (2024). DE-COP: Detecting Copyrighted Content in Language Models Training Data. ArXiv, abs/2402.09910.
> > > > > > > > >
> > > > > > > > > [7] Soldaini, L., Kinney, R., Bhagia, A., Schwenk, D., Atkinson, D., Authur, R., Bogin, B., Chandu, K.R., Dumas, J., Elazar, Y., Hofmann, V., Jha, A., Kumar, S., Lucy, L., Lyu, X., Lambert, N., Magnusson, I., Morrison, J.D., Muennighoff, N., Naik, A., Nam, C., Peters, M.E., Ravichander, A., Richardson, K., Shen, Z., Strubell, E., Subramani, N., Tafjord, O., Walsh, P., Zettlemoyer, L.S., Smith, N.A., Hajishirzi, H., Beltagy, I., Groeneveld, D., Dodge, J., & Lo, K. (2024). Dolma: an Open Corpus of Three Trillion Tokens for Language Model Pretraining Research. ArXiv, abs/2402.00159.
> > > > > > > > >
> > > > > > > > > [8] Chang, K.K., Cramer, M.H., Soni, S., & Bamman, D. (2023). Speak, Memory: An Archaeology of Books Known to ChatGPT/GPT-4. ArXiv, abs/2305.00118.
> > > > > > > > >
> > > > > > > > > [9] Shi, W., Ajith, A., Xia, M., Huang, Y., Liu, D., Blevins, T., Chen, D., & Zettlemoyer, L.S. (2023). Detecting Pretraining Data from Large Language Models. ArXiv, abs/2310.16789.
> > > > > > > > >
> > > > > > > > > [10] Fu, W., Wang, H., Gao, C., Liu, G., Li, Y., & Jiang, T. (2023). Practical Membership Inference Attacks against Fine-tuned Large Language Models via Self-prompt Calibration. ArXiv, abs/2311.06062.
> > > > > > > > >
> > > > > > > > > [11] Mattern, J., Mireshghallah, F., Jin, Z., Scholkopf, B., Sachan, M., & Berg-Kirkpatrick, T. (2023). Membership Inference Attacks against Language Models via Neighbourhood Comparison. ArXiv, abs/2305.18462.
> > > > > > > > >
> > > > > > > > > [12] Tran, T., Liu, R., & Xiong, L. (2025). Tokens for Learning, Tokens for Unlearning: Mitigating Membership Inference Attacks in Large Language Models via Dual-Purpose Training. ArXiv, abs/2502.19726.
> > > > > > > > >
> > > > > > > > > [13] Naseh, A., Peng, Y., Suri, A., Chaudhari, H., Oprea, A., & Houmansadr, A. (2025). Riddle Me This! Stealthy Membership Inference for Retrieval-Augmented Generation. ArXiv, abs/2502.00306.
> > > > > > > > >
> > > > > > > > > [15] Zhang, R., Bertran, M., & Roth, A. (2024). Order of Magnitude Speedups for LLM Membership Inference. ArXiv, abs/2409.14513.
> > > > > > > > >
> > > > > > > > > [16] Chang, H., Shamsabadi, A.S., Katevas, K., Haddadi, H., & Shokri, R. (2024). Context-Aware Membership Inference Attacks against Pre-trained Large Language Models. ArXiv, abs/2409.13745.[14] Kim, G., Li, Y., Spiliopoulou, E., Ma, J., Ballesteros, M., & Wang, W.Y. (2024). Detecting Training Data of Large Language Models via Expectation Maximization. ArXiv, abs/2410.07582.
> > > > > > > > >
> > > > > > > > > [17] Xie, R., Wang, J., Huang, R., Zhang, M., Ge, R., Pei, J., Gong, N.Z., & Dhingra, B. (2024). ReCaLL: Membership Inference via Relative Conditional Log-Likelihoods. Conference on Empirical Methods in Natural Language Processing.

---

> > > > > > > > > > ### Author Response · Authors · 2025-06-05
> > > > > > > > > > **Rebuttal Follow-Up**
> > > > > > > > > >
> > > > > > > > > > Thank you again for your time and valuable feedback on our paper.
> > > > > > > > > >
> > > > > > > > > > We wanted to gently follow up on our rebuttal. We are eager to know if our responses have addressed your concerns and would be happy to provide any further clarifications.
> > > > > > > > > >
> > > > > > > > > > Please let us know if you have any follow-up questions.

---

> > > > > > > > > > > ### Comment · Reviewer_Yujw · 2025-06-07
> > > > > > > > > > >
> > > > > > > > > > > Thank you for your rebuttal. I've revised my score

---

> > > > > > > > > > > > ### Author Response · Authors · 2025-06-10
> > > > > > > > > > > >
> > > > > > > > > > > > Thank you for your thoughtful feedback and for raising your score! We're glad our updates addressed your concerns, and we truly appreciate your support in strengthening the paper.

---

### Decision · Program_Chairs · 2025-07-08

**Decision:**

Accept

**Comment:**

This paper proposes a very simple membership inference attack in which n-gram overlap metrics are computed on multiple completions of a prefix. The output requires access to only the token outputs of the model and works better as more completions are generated, and both open and closed models are tested (the latter of course with "silver-standard" assumptions instead of the unknowable ground truth as stated by the authors). One major concern with the paper was that its evaluation was not conducted in low-FPR regimes exclusively, but the authors added results on this in their rebuttal that showed strong performance compared to baselines here. There are still concerns with comprehensiveness in the related work section, as well as some overclaiming in the writing noted by reviewers. Overall, though, it is a solid contribution to LLM membership inference research and should be used as a baseline in all future work in this area.